# Predicting lapses of attention with sleep-like slow waves

Thomas Andrillon [1,2 ✉], Angus Burns [1], Teigane Mackay [1], Jennifer Windt[3] & Naotsugu Tsuchiya [1,4,5]

Attentional lapses occur commonly and are associated with mind wandering, where focus is turned to thoughts unrelated to ongoing tasks and environmental demands, or mind blanking, where the stream of consciousness itself comes to a halt. To understand the neural mechanisms underlying attentional lapses, we studied the behaviour, subjective experience and neural activity of healthy participants performing a task. Random interruptions prompted participants to indicate their mental states as task-focused, mind-wandering or mind-blanking. Using high-density electroencephalography, we report here that spatially and temporally localized slow waves, a pattern of neural activity characteristic of the transition toward sleep, accompany behavioural markers of lapses and preceded reports of mind wandering and mind blanking. The location of slow waves could distinguish between sluggish and impulsive behaviours, and between mind wandering and mind blanking. Our results suggest attentional lapses share a common physiological origin: the emergence of local sleep-like activity within the awake brain.

[1] School of Psychological Sciences, Turner Institute for Brain and Mental Health, Monash University, Melbourne, VIC, Australia. [2] Institut du Cerveau—Paris Brain Institute—ICM, Sorbonne Université, Inserm, CNRS, Paris, France. [3] Philosophy Department, Monash University, Melbourne, VIC, Australia. [4] Center for Information and Neural Networks (CiNet), National Institute of Information and Communications Technology (NICT), Suita, Osaka, Japan. [5] Advanced Telecommunications Research Computational Neuroscience Laboratories, Soraku-gun, Kyoto, Japan. ✉email: thomas.andrillon@icm-institute.org

The human brain sustains the stream of our conscious experiences. Attention can direct cognitive resources toward the external world and enable the selection and amplification of information relevant to an individual's current behavioural goals[1]. But attention can also turn inward, as is the case when we focus on internally generated task-unrelated thoughts, a phenomenon usually referred to as mind wandering (MW)[2]. Recent investigations have also shown that the stream of thoughts can come to a pause when individuals who are awake are left with the feeling of an empty mind (mind blanking (MB))[3].

Shifts of attention towards the internal world, invoking MB and MW, can occur spontaneously without our knowledge or will[4]. In fact, a characteristic feature of attention is its fleeting nature and the difficulty to maintain it on a task for long periods of time[1,5]. In this paper, we define lapses of attention as the shift of the focus of thoughts away from the task at hand or environmental demands. The consequences of attentional lapses are very diverse. At the behavioural level, they can result in a lack of responsiveness or sluggish reactions, but they can also result in impulsive responses[6]. Curiously, these behavioural failures can be accompanied by a lack of conscious awareness and the absence of mental activity (MB[3]), or rich, spontaneous mental activity (MW[2]).

It is yet unclear whether these different types of attentional lapses (sluggish vs. impulsive behaviours; MB vs. MW) belong to a disparate family of behavioural and phenomenological events[7,8], each of them associated with different physiological causes[9,10], or whether they can be traced back to common underlying physiological causes[11]. Previous models of MW have proposed that MW and MB might arise in distinct neurophysiological states[3,9,10]. However, the fact that both sluggish and impulsive responses increase following sleep deprivation[12,13] and in individuals with attentional deficits[6,14] implies a common mechanism. Likewise, sleepiness has been associated with both MW and MB[15,16] despite these two mental states being phenomenologically distinct[3]. Furthermore, investigations of the sleep onset period (hypnagogia) indicate that subjective experiences resembling MW (focus on internally generated contents) and MB (loss of awareness) can both occur at the border between wakefulness and sleep[17,18]. Interestingly, these studies seem to associate lapses with pressure for sleep, suggesting an involvement of fatigue.

Indeed, each hour spent awake comes at the cost of mounting sleep pressure. Past research suggests that the need for sleep might only be dissipated by sleep itself[19], as sleep plays a vital role in neural homeostasis[20]. When individuals are prevented from sleeping for extended periods of time (as in sleep deprivation studies), a subset of brain regions can start displaying electroencephalographic (EEG) signatures of non-rapid eye-movement (NREM) sleep in the form of sleep-like slow waves (within the delta (1–4 Hz) or theta (4–7 Hz) range), despite individuals being behaviourally and physiologically awake[21,22]. These sleep-like slow waves within wakefulness are referred to as local sleep in contrast with the global whole-brain transition commonly observed at sleep onset[22–25]. Importantly, both local and global transitions toward sleep are characterized by the same neural signature: the occurrence of high-amplitude slow waves.

It has been proposed that the multiplication of sleep-like slow waves could perturb brain functions and cause behavioural lapses[22]. In fact, during sleep, slow waves are associated with episodes of widespread neural silencing[26], which have been connected to sensory disconnection, behavioural unresponsiveness, and the loss of consciousness[27,28]. Intracranial studies in humans and rodents showed that slow waves during waking are associated with reduced neuronal firing, resulting in behavioural errors[21,22]. Wake slow waves can also be detected in human non-invasive recordings[29–31] and here again the amount of slow waves recorded in a given brain region correlates with the number of errors performed in a task recruiting this specific brain region[29,30]. These results strongly suggest that slow waves could explain the behavioural component of attentional lapses[22]. However, the impact of slow waves on conscious experience, including MW and MB, is still unclear.

In a recent review, we proposed that local sleep, manifested as the occurrence of slow waves in waking, could not only explain the behavioural consequences of attentional lapses, both regarding sluggish and impulsive responses, but also their phenomenology, like MW or MB[11]. We also argued that local sleep is not an extreme phenomenon, occurring only when individuals are pushed to their limit, but could occur in well-rested individuals[31] and explain the occurrence of lapses in our everyday lives. To test this framework, we rely on the detection of slow waves and formulate three different hypotheses as follows: (i) Can slow waves predict, at the single-trial level, both sluggish and impulsive behaviours in well-rested individuals? (ii) Are slow waves associated with both MW and MB? (iii) Does the location of slow waves (i.e. which electrodes show slow waves) differentiate between sluggishness and impulsivity, MB and MW? Through these hypotheses we will test the idea that local slow waves could act as a functional switch, transiently perturbing the functioning of a given cortical network. Accordingly, a common physiological event (local slow waves) could lead to drastically different outcomes depending on its location within the brain.

In this work, we cross-examined the behavioural performance, subjective reports and physiological data from healthy individuals ($N = 26$) performing undemanding Go/NoGo tasks (Fig. 1a). We sampled participants' subjective experience by interrupting them during the task and asking them a series of questions about their mental states prior to the interruption, including whether they were focusing on the task, MW or MB (Fig. 1b). Finally, we recorded their brain activity using high-density scalp EEG and pupil size as objective proxies for participants' level of vigilance. Importantly, here participants were neither sleep-deprived nor placed in conditions favouring dozing off (i.e. not in bed or reclined, task requiring constant attention and responses).

## Results

**Task performance and subjective experience.** The Go/NoGo tests (see "Methods") require participants' sustained attention, but our participants declared focusing on the task only in ~48% of the probes (Face Task: 49.4 ± 4.9%; Digit Task: 47.2 ± 5.1%; mean ± standard error of the mean (SEM) across $N = 26$ participants; Fig. 1c). The rest of the time, they declared thinking about something else (MW; Face: 38.0 ± 4.3%; Digit: 40.9 ± 4.8%; Fig. 1c) or thinking about nothing (MB; Face: 12.7 ± 3.0%; Digit: 11.9 ± 2.9%; Fig. 1c). These results are well in line with previous findings[32,33] and highlight the prevalence of attentional lapses. Attentional lapses were also reflected in participants' accuracy on the Go trials (Face: 3.1 ± 0.4% of miss, i.e., errors on Go trials; Digit: 1.7 ± 0.2%; Fig. 2a), but more notably on the NoGo trials (Face: 35.0 ± 2.5% of false alarms (FA), i.e., errors on NoGo trials; Digit: 32.5 ± 2.7%; Fig. 2b).

Next, we focused on the behavioural patterns preceding subjective reports of attentional lapses. Specifically, we examined participants' behaviour in the 20 s before the onset of the probes that led to MW and MB reports (see "Methods"). To quantify the impact of mental states (i.e. ON, MW or MB) on behaviour, we compared statistical models that either did or did not include mental states as a predictor of behaviour (see "Methods" and Supplementary Table 1). A significantly better fit by the model incorporating mental states, assessed through a likelihood ratio test, was interpreted as evidence for the influence of mental states.

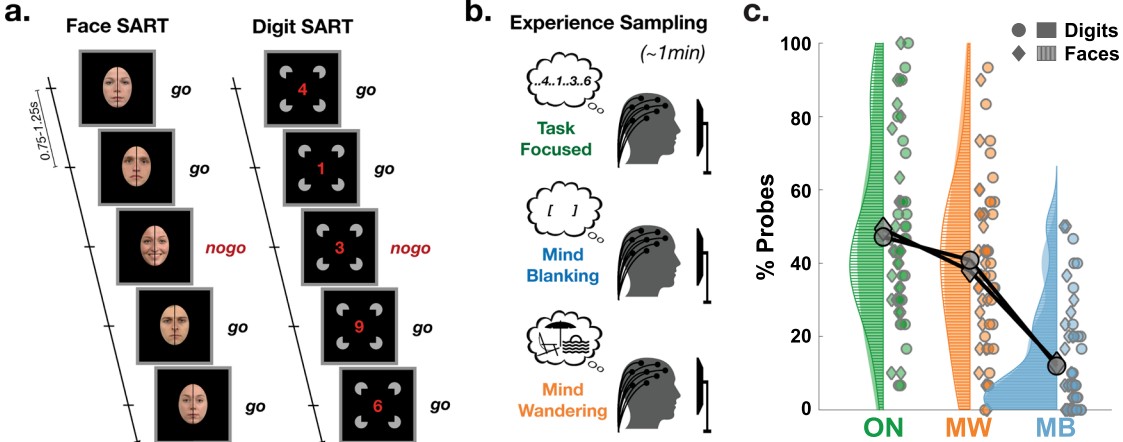

**Fig. 1 Experimental design and hypotheses. a** Participants performed both a SART (Sustained Attention to Response Task) on faces stimuli (NoGo trials: smiling faces) and a SART on digits (NoGo trials: 3). Face/Digit presentation was continuous (new face/digit every 0.75–1.25 s). Images of faces were obtained from the Radboud Face Database (see "Methods"). **b** Every 30 to 70 s, participants were interrupted and instructed to report their mental state (see "Methods" and Supplementary Methods). Most importantly, they were asked to indicate whether they were focusing on the task (task-focused: ON), thinking about nothing (mind blanking: MB) or thinking about something other than the task (mind wandering: MW). High-density EEG and pupil size were continuously recorded throughout the task. **c** Proportion of mental states reported during probes categorized as task-focused (ON, green), mind wandering (MW, orange) and mind blanking (MB, blue) during the tasks with Digits (circles for each individual participant; filled surfaces for smoothed density plot) and Faces (diamonds and surfaces with horizontal stripes). Grey diamonds and circles show the average across participants.

To describe the size and direction of the statistical effects, we report the estimates ($\beta$) of the contrasts of interest (MW vs. ON, MB vs. ON and MB vs. MW) and their 95% confidence interval (CI). Accordingly, we found a significant effect of mental states on misses (model comparison: $\chi^2(2) = 36.0$, $p = 1.5 \times 10^{-8}$; Fig. 2a). Specifically, MW and MB were associated with an increase in misses compared to ON reports (MW vs. ON: $\beta = 0.011$, CI: [0.005, 0.016]; MB vs. ON: $\beta = 0.023$, CI: [0.015, 0.032]) and misses were more frequent for MB compared to MW reports (MB vs. MW: $\beta = 0.013$, CI: [0.005, 0.021]). FAs were also modulated across mental states (model comparison: $\chi^2(2) = 115.9$, $p < 10^{-16}$; see Fig. 2b), with an increase for both MW and MB compared to ON (MW vs. ON: $\beta = 0.21$, CI: [0.17, 0.24]; MB vs. ON: $\beta = 0.17$, CI: [0.12, 0.23]), but similar levels of FAs for MB and MW (MB vs. MW: $\beta = -0.028$, CI: [-0.084, 0.028]). Finally, mental states were associated with different patterns of reaction times (RTs; model comparison: $\chi^2(2) = 16.3$, $p = 2.9 \times 10^{-4}$; Fig. 2c) with slower RTs for MB compared to both ON and MW reports (MB vs. ON: $\beta = 0.019$, CI: [0.009, 0.030]; MB vs. MW: $\beta = 0.022$, CI: [0.011, 0.032]; MW vs. ON: $\beta = -0.0025$, CI: [-0.0096, 0.0045]).

Taken together, these results suggest that MW and MB decrease performance in different ways. MB is characterized by more misses and slower RT than MW and ON, which is consistent with the idea that MB induces sluggish mental states. As to MW, the analysis of misses may be ostensibly compatible with an idea of MW as a mild form of MB; however, the fact that RT in MW is faster than that in MB is not easy to reconcile with such an idea. Instead, this overall pattern is consistent with an idea that MW induces more impulsive mental states.

The same pattern of results was obtained when normalizing misses, FAs and RT in the MW and MB conditions by the average obtained in the ON condition for each subject (see Supplementary Table 2). We also obtained qualitatively the same results when focusing on a shorter time window (10 s before probe onsets instead of 20 s; see Supplementary Table 3).

**Vigilance.** Although MW and MB differ according to their phenomenological definition and associated behaviours, both states seem to occur in a similar context of low vigilance (as reported by participants themselves during probes). To address this, we quantified the degrees of correlation between participants' vigilance ratings and each mental state (comparison between models including or not the information about the mental state: $\chi^2(2) = 144.8$, $p < 10^{-16}$; Fig. 2d and Supplementary Table 1). Participants reported lower vigilance ratings for both MW and MB compared to ON (MW vs. ON: $\beta = -0.39$, CI: [-0.40, -0.37] and MB vs. ON: $\beta = -0.53$, CI: [-0.55, -0.50]). Vigilance ratings were even lower for MB compared to MW (MB vs. MW: $\beta = -0.13$, CI: [-0.24, -0.02]). We then examined a classical objective proxy for vigilance: pupil size[34,35]. Pupil size prior to probes (Fig. 2e, N = 25 participants here, see "Methods") was significantly modulated by mental states (model comparison: $\chi^2(2) = 18.0$, $p = 1.2 \times 10^{-4}$) with MW and MB associated with smaller pupils than ON probes (MW vs. ON: $\beta = -0.29$, CI: [-0.43, -0.15]; MB vs. ON: $\beta = -0.22$, CI: [-0.45, -0.003]). Pupil size did not differ between MW and MB (MB vs. MW: $\beta = 0.065$, CI: [-0.16, 0.29]), despite the significant correlation between vigilance ratings and pupil size (model comparison: $\chi^2(2) = 134.5$, $p < 10^{-16}$; see Supplementary Table 1 for details on the models). This implies that these two measures of vigilance can be differentially sensitive to mental states.

**Slow waves.** To further examine the potential mechanistic link between sleep pressure and attentional lapses, we set out to detect a marker of sleep pressure in the EEG signal, in the form of local, sleep-like slow waves. We recently reviewed the rationale behind this approach[11]. In particular, relying on a local marker of sleep allowed us to test whether distinct families of attentional lapses can be coherently explained by the occurrence and spatio-temporal characteristics of slow waves.

Following evidence from refs. [21,22,29,30], we operationally defined slow waves as the occurrence of large-amplitude waves within the delta ([1–4] Hz) range. We first detected the occurrence of slow waves in each EEG electrode using an established approach developed in wakefulness and sleep (see refs. [30,31,36] and "Methods"). Both the temporal profile and topographical distribution of slow waves detected during the tasks

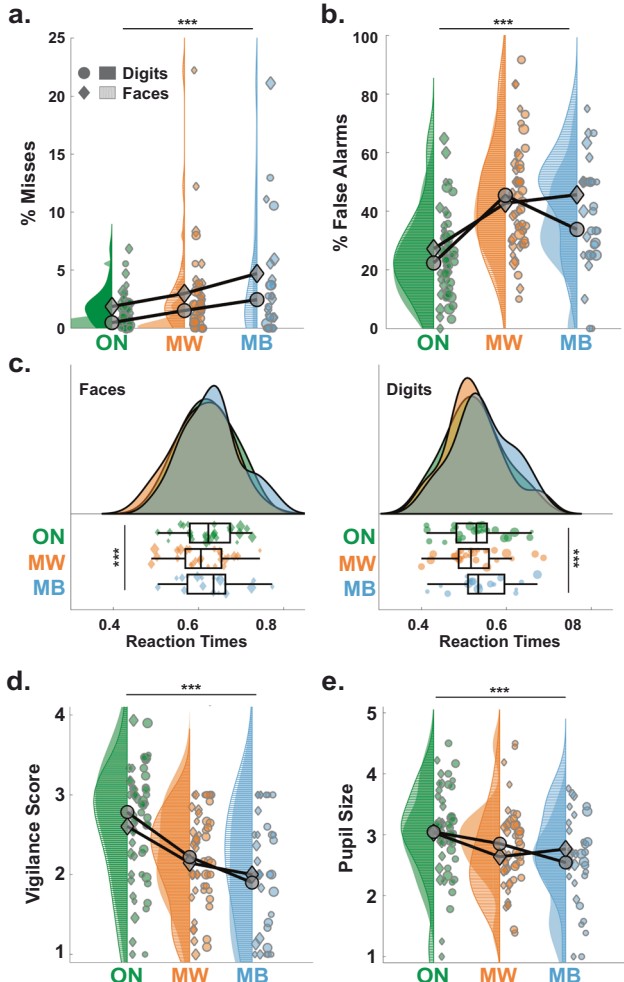

**Fig. 2 Low arousal is associated with attentional lapses characterized by different behavioural outcomes.** Proportion of misses (**a**) and false alarms (**b**) in the 20 s preceding task-focused (ON, green), mind wandering (MW, orange) and mind blanking (MB, blue) during the tasks with Digits (circles for each individual participant; filled surfaces for smoothed density plot) and Faces (diamonds and surfaces with horizontal stripes). The markers' size is proportional to the number of reports for each participant (same for **c–e**). Grey diamonds and circles show the average across participants, weighted by the number of reports (same for **c–e**). **c** Distribution of reaction times (RT) for Go Trials (left: Face; right: Digit) in the 20 s preceding ON, MW and MB reports. Box plots show the mean (central bar), the lowest and highest individual data points (end of the whiskers) and the lower and higher quartiles (edges of the box). On top of each plot are shown the smoothed density plot for the different mental states. **d** Vigilance scores (subjective ratings provided during probes) associated with ON, MW and MB reports. **e** Discretized pupil size (see "Methods") in the 20 s preceding ON, MW and MB reports. In **a–e**, stars show the level of significance of the effect of mental states (likelihood ratio test, see "Methods"; ***$p < 0.005$; **a**: $p = 1.5 \times 10^{-8}$; **b**: $p < 2 \times 10^{-16}$; **c**: $p = 2.2 \times 10^{-4}$; **d**: $p < 2 \times 10^{-16}$; **e**: $p = 1.2 \times 10^{-4}$). $N = 26$ except for panel (**e**), where $N = 25$.

(Fig. 3a, b) resemble the slow waves observed in NREM sleep[36,37]. This is not trivial as our detection algorithm did not select this specific temporal profile or topographical distributions.

*Relationship between global properties of slow waves and vigilance.* Next, we checked whether global properties of the slow waves index participants' level of vigilance. We correlated first participants' subjective vigilance ratings with the amount and properties of slow

waves. The amount and properties of slow waves were extracted prior to probe onset ([−20, 0] s) and averaged across the entire scalp (all 63 electrodes, see "Methods"). Model comparisons between models with or without slow-wave density, amplitude, downward slope or upward slope indicate a negative correlation between vigilance ratings and slow-wave density ($\chi^2(1) = 13.1$, $p = 3.9 \times 10^{-4}$, $\beta = -0.074$, CI: [−0.114, −0.034]), amplitude ($\chi^2(1) = 33.1$, $p = 8.5 \times 10^{-9}$, $\beta = -0.023$, CI: [−0.031, −0.015]), downward slope ($\chi^2(1) = 82.1$, $p < 10^{-16}$, $\beta = -2.5 \times 10^{-3}$, CI: [−3.1 × 10$^{-3}$, −2.0 × 10$^{-3}$]), and upward slope ($\chi^2(1) = 47.3$, $p < 10^{-11}$, $\beta = -1.7 \times 10^{-3}$, CI: [−2.2 × 10$^{-3}$, −1.2 × 10$^{-3}$]) (see also Supplementary Fig. 1). We then verified that slow-wave properties were negatively correlated with pupil size recorded before each probe (model comparison between models with or without slow-wave density, amplitude, downward slope or upward slope: density: $\chi^2(1) = 12.3$, $p = 4.6 \times 10^{-4}$, $\beta = -0.13$, CI: [−0.21, −0.059]; amplitude: $\chi^2(1) = 7.4$, $p = 6.4 \times 10^{-3}$, $\beta = -0.0088$, CI: [−0.015, −0.0025]; downward slope: $\chi^2(1) = 15.1$, $p = 1.0 \times 10^{-4}$, $\beta = -6.9 \times 10^{-4}$, CI: [−1.0 × 10$^{-3}$, −3.4 × 10$^{-4}$]; upward slope: $\chi^2(1) = 17.6$, $p = 2.7 \times 10^{-5}$, $\beta = -7.6 \times 10^{-4}$, CI: [−1.1 × 10$^{-3}$, −4.0 × 10$^{-4}$]) (see also Supplementary Fig. 2). Finally, we confirmed that the amount of slow waves detected increased with time spent on task (Supplementary Fig. 3), as can be expected when considering the homeostatic regulation of sleep and local sleep[22].

*Relationship between local properties of slow waves and mental states at the probe level.* Local properties of the slow waves (Fig. 3c, d) predicted reports of mental states. Specifically, we examined, for each scalp electrode, whether slow-wave properties (density, amplitude, downward and upward slopes) prior to probes were predictive of the mental state reported by participants. To do so, we focused on pairwise contrasts (MW vs. ON, MB vs. ON and MB vs. MW) and performed this analysis for each electrode independently. A cluster-permutation approach ($p_{cluster} < 0.05$, Bonferroni-corrected cluster threshold; see "Methods") revealed that MW (compared to ON) was predicted by an increase in the number of slow waves and slow-wave amplitude over frontal electrodes, and by an increase in slow waves' downward and upward slope over centro-frontal electrodes (Fig. 4a). MB (compared to ON; Fig. 4b) was also predicted by an increase in slow-wave density over frontal areas as well as by the slow-wave slopes (downward and upward) over centro-parietal electrodes (no significant cluster for slow-wave amplitude). Finally, a direct contrast between MB and MW (Fig. 4c) indicated that a reduction of slow-wave amplitude over frontal electrodes but an increase in their upward slope over parietal electrodes were predictive of MB. No significant clusters were obtained for slow-wave density and downward slope.

To provide more details on the temporal relationship between slow waves and mental states, we replicated this analysis by splitting the 20 s window before probes into four windows of 5 s (Supplementary Figs. 4–6). The contrasts between MW and ON (Supplementary Fig. 4) as well as MB and ON (Supplementary Fig. 5) show that the properties of slow waves best predict mental states within the 5 s before a probe. In terms of topography, when compared to ON state, MW is best predicted by the slow-wave properties in frontal electrodes (Supplementary Fig. 4), while MB is best predicted by those in the centro-parietal electrodes (Supplementary Fig. 5).

*Relationship between local properties of slow waves and behaviour at the trial level.* To further understand the association between slow waves and attentional lapses, we examined the influence of slow waves on participants' behaviour at the single-trial level (for all trials within 20 s of a probe onset, independently of mental

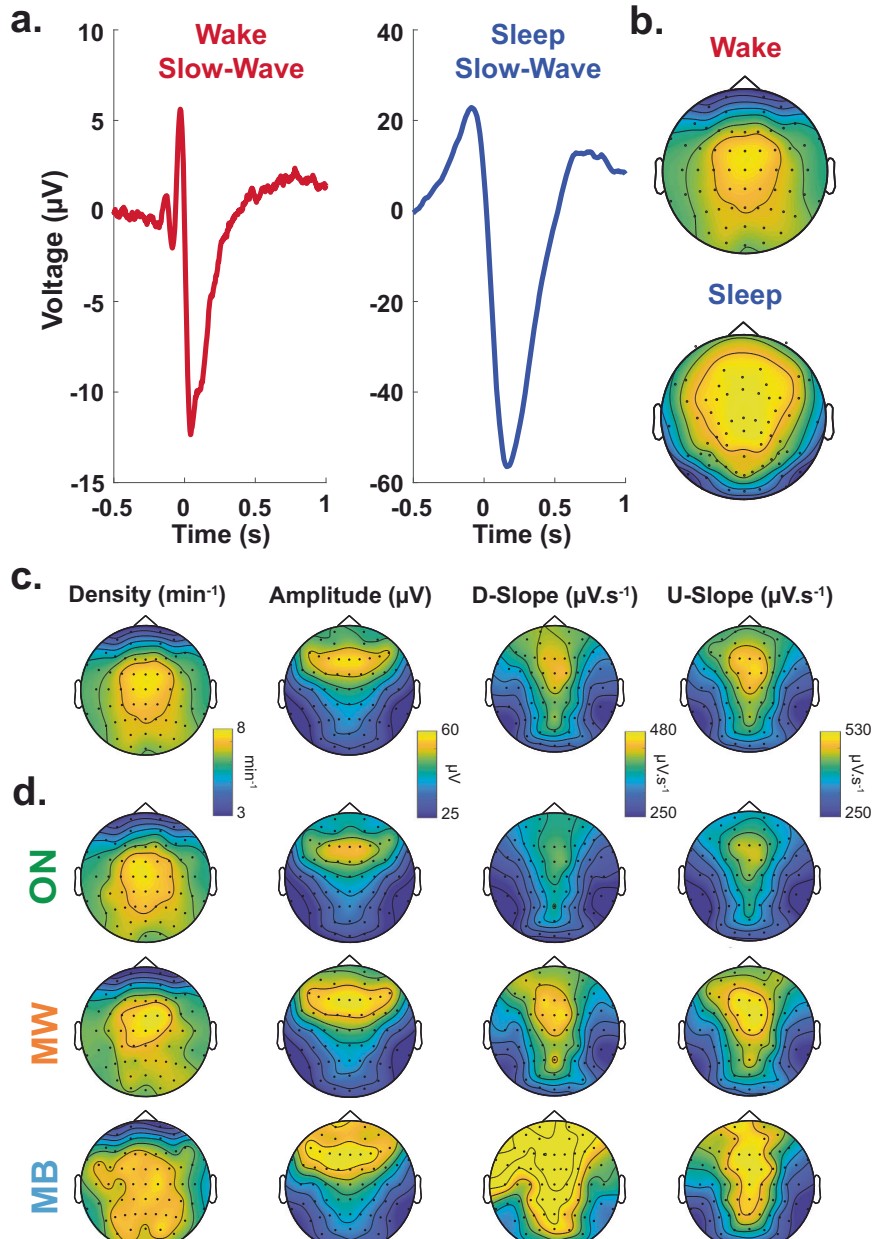

**Fig. 3 Properties of slow waves. a** Average waveform of the slow waves detected over electrode Cz during the behavioural tasks (red, left; $N = 26$ participants). The average waveform of slow waves detected during sleep (blue, right), extracted from another dataset (see Supplementary Methods), is shown for comparison. Slow waves were aligned by their start, defined as the first zero-crossing before the negative peak (see "Methods"). **b** Scalp topographies of the density of slow waves (arbitrary units) detected in wakefulness (top) and sleep (bottom). **c** Scalp topographies of wake slow waves properties (first column: temporal density; second: peak-to-peak amplitude; third: downward slope (D-slope); fourth: upward slope (U-slope); see "Methods") averaged across participants ($N = 26$). **d** Scalp topographies for slow-wave density (first column), amplitude (second), downward slope (D-slope, third) and upward slope (U-slope, fourth) for the different mental state (ON (task-focused), MW (mind wandering) and MB (mind blanking)).

states). To do so, for each trial and electrode, we marked the presence or absence of slow waves between stimulus onset and offset (see "Methods") and used this as a (binary) predictor of RT, misses and FAs (Fig. 5). This analysis revealed spatially specific effects of slow waves on distinct behavioural outcomes. Namely, slow waves in frontal electrodes co-occurred with faster RTs, while slow waves in posterior electrodes co-occurred with slower RTs (Fig. 5a, $p_{\text{cluster}} < 0.05$, Bonferroni-corrected cluster threshold). Likewise, frontal slow waves were associated with more FAs (a marker of impulsivity, Fig. 5b) and posterior slow waves with more misses (a marker of sluggishness, Fig. 5c).

**Decision modelling**. Finally, we implemented an influential model of two-alternative forced-choice (2AFC) decision-making: the diffusion decision model (DDM)[38]. The DDM decomposes full RT distributions and choice proportions into latent cognitive processes that are thought to underlie participants' decisions in 2AFC tasks (see "Methods" and Supplementary Methods and Supplementary Fig. 7). These include the time it takes for participants to start computing their response (non-decision time or NDT [$t$]), the speed at which participants accumulate evidence for the two responses (drift rates for Go [$v_{\text{Go}}$] and NoGo [$v_{\text{NoGo}}$] responses), the amount of evidence needed to reach a decision (decision

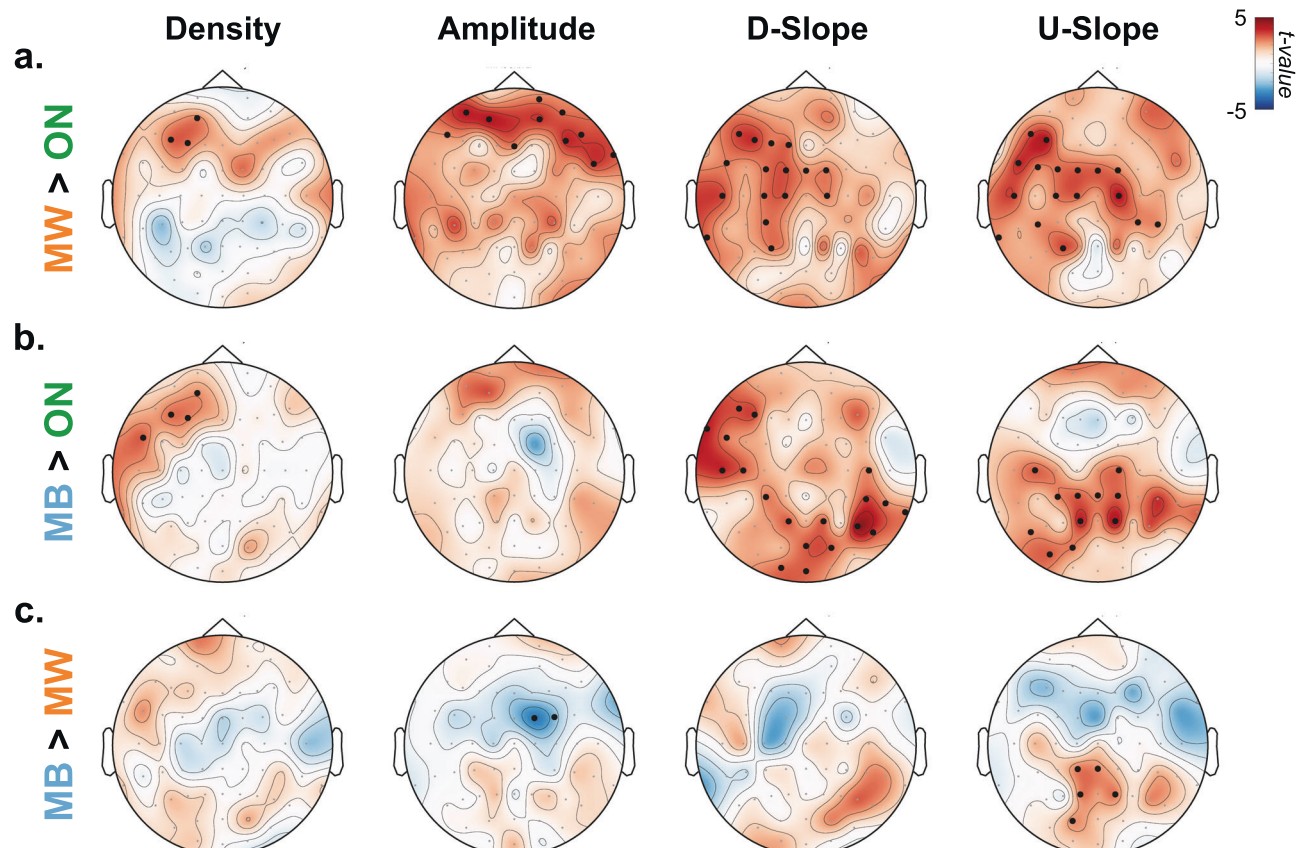

**Fig. 4 Local properties of slow waves are predictive of mental states.** Locally, based on each individual electrode, we performed mixed-effect analyses, following with permutation analysis to quantify the impact of slow-wave properties on mental states. First column: density; second: amplitude; third: downward slope (D-slope); fourth: upward Slope (U-slope). These slow-wave properties were extracted for each electrode and used to compute the $t$ values (shown in each topographical plot) from mixed-effect models on the following comparisons: **a** MW > ON, **b** MB > ON and **c** MB > MW ($N = 26$ participants). Black dots denote significant clusters of electrodes ($p_{cluster}$ <0.05 corrected for 12 comparisons using a Bonferroni approach, see "Methods"). ON task-focused, MW mind wandering, MB mind blanking.

threshold [$a$]), the participants' initial bias for one of the two responses (decision bias [$z$]) and the bias for the accumulation of evidence for one of the two responses (drift bias [$v_{Bias}$]). A hierarchical Bayesian approach was used to fit the DDM to the RTs obtained in the Go/NoGo tests[39] so that each parameter ($v_{Go}$, $v_{NoGo}$, $a$, $z$, $t$ and $v_{Bias}$) was free to vary by participant, task and mental states (ON, MW and MB) or slow-wave occurrence (present vs. absent). Simulations confirmed that this hierarchical DDM can successfully predict the observed data (Supplementary Fig. 8).

Considering the trials that were within 20 s from the onset of the probes, we first estimated these different parameters of the DDM for each mental state (Supplementary Fig. 9). This analysis shows differences between MW and MB reports in terms of decision bias and threshold (Supplementary Fig. 9). The lower threshold and higher decision bias observed for MW (compared to MB) are concordant with the idea that MW is associated with behavioural impulsivity.

We then used this modelling approach to examine how the occurrences of slow waves impact the different cognitive processes leading to participants' responses, with a particular focus being the test of our core hypothesis: the presence of slow waves in frontal areas leads to impulsivity by disrupting the cognitive mechanisms underlying executive control, while the presence of slow waves in posterior areas leads to sluggishness by slowing down the integration of sensory inputs.

We report here the differences in the parameters' estimates in the presence or absence of slow waves detected for each electrode

(Fig. 6). The associated scalp topographies indicate both global and local (electrode-specific) effects of slow waves on DDM parameters. As global effects (Fig. 6d–f), we found first that the presence of slow waves was associated with a reduction in decision threshold ($a$; Fig. 6d), consistent with the idea that slow waves facilitate impulsive responses. Second, slow waves were also associated with longer NDTs ($t$; Fig. 6e), suggesting that slow waves can slow down neural processes underlying stimulus encoding and/or motor preparation. Finally, the presence of slow waves was correlated with an increase in the starting point of the decision process (prior bias $z$; Fig. 6f), implying shifts in the decision process towards Go responses.

As local electrode-specific effects, we observed again contrasting results between posterior and frontal slow waves (Fig. 6a–c). Slow waves within posterior electrodes were associated with a reduction of $v_{Go}$ and $v_{Bias}$, meaning that evidence accumulation was slower for Go decisions and the drift bias of the decision process for Go responses was reduced (Fig. 6a, c). Conversely, slow waves within frontal electrodes correlated with a reduction of $v_{NoGo}$, indicating that evidence accumulation was slower for NoGo Decisions (Fig. 6b). These modelling results provide a potential explanation of how local slow-wave properties relate to mental states (Fig. 4) and single-trial task performance (Fig. 5). In other words, posterior slow waves were associated with sluggish responses and increased misses, while frontal slow waves were associated with faster, impulsive responses and more FAs. Indeed, slower evidence accumulation in favour of Go responses would

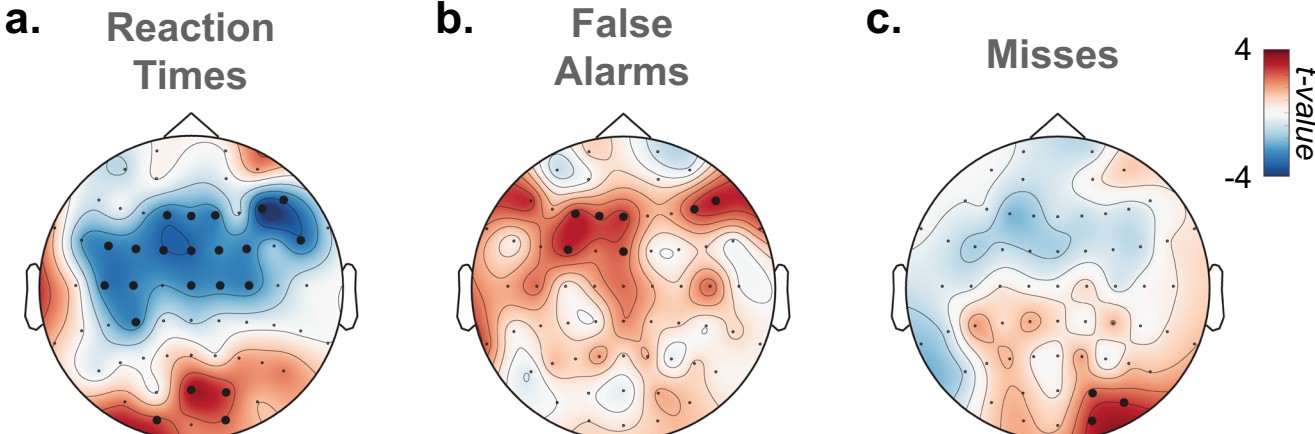

**Fig. 5 Local occurrences of slow waves are associated with modulations of performance.** Mixed-effects models were used to quantify the correlation between slow-wave occurrence and reaction times (**a**), false alarms (**b**) and misses (**c**) at the single-trial level. Topographies show the scalp distribution of the associated $t$ values ($N = 26$ participants). Black dots denote significant clusters of electrodes ($p_{cluster}$ <0.05 corrected for three comparisons using a Bonferroni approach, see "Methods").

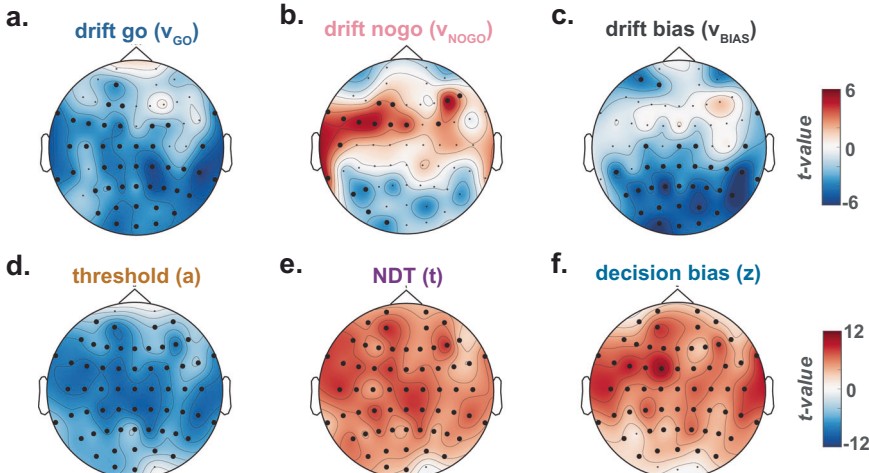

**Fig. 6 Global and local effects of the occurrence of slow waves on sub-components of decision-making.** Reaction times in the Go/NoGo tasks were modelled according to a Hierarchical Drift-Diffusion Model (see "Methods"). **a–f** Topographical maps of the effect of slow waves (i.e. whether or not a slow wave was detected for each trial and for a specific electrode) on the parameters of decision-making: drift Go [$v_{Go}$] (**a**), drift NoGo [$v_{NoGo}$] (**b**), drift bias [$v_{Bias}$] (**c**), threshold [$a$] (**d**), non-decision time or NDT [$t$] (**e**), decision bias [$z$] (**f**). The effect of slow-wave occurrence was estimated with LMEs (see "Methods") and topographies show the scalp distribution of the associated $t$ values ($N = 26$ participants). Black dots denote significant clusters of electrodes ($p_{cluster}$<0.05, Bonferroni-corrected, see "Methods").

lead to slower RTs or even misses, whereas slower evidence accumulation for NoGo responses would lead to faster responses and possibly FAs.

These results suggest that slow waves could represent both a general index of fatigue and a mechanism underlying specific behavioural and subjective consequences of sleepiness. In other words, regardless of where it happens, slow waves can reflect the global sleepiness of an individual. Meanwhile, at a finer grain, depending on where they occur, slow waves can disrupt specific cognitive processes carried out by the affected brain regions. Taken together, we interpret this as strong evidence in support of the idea that slow waves are a compelling physiological phenomenon, which precedes and co-occurs with subjective and behavioural aspects of attentional lapses. In addition, our results indicate that spatio-temporal properties of slow waves can specifically explain distinctive components of behavioural and mental consequences of attentional lapses in a unified and quantitative manner.

## Discussion

According to both in-lab and real-life studies, humans spend up to half of their waking lives not paying attention to their environment or any task at hand[32,33]. However, the ubiquitousness of these attentional lapses remains unexplained. Part of the difficulty in identifying their neural correlates could be due to their intrinsic diversity[7,8]. We embraced this challenge to clarify the neural mechanisms underlying attentional lapses by linking three different levels of explanation: behaviour, phenomenology and physiology. To grasp the phenomenological diversity of attentional lapses, we distinguish between MW and MB, where MW is operationally defined as a shift in attention away from environmental demands and toward spontaneous task-unrelated thoughts, and MB as the absence of thoughts and an empty mind. We propose that these two types of attentional lapses can be explained by the occurrence of sleep-like, low-frequency, high-amplitude waves, where regional differences in the occurrence of slow waves predict the type of attentional lapse as well as its

behavioural profile. These slow waves have been previously linked to the transition toward sleep when they are visible across the scalp[37] or to the concept of local sleep when they occur in a subset of electrodes[11,22,25,29,30] and in the context of an awake individual.

The concept of local sleep builds upon a recent questioning of the classical view of sleep as an all-or-nothing phenomenon[22–24,40,41]. Although sleep is orchestrated at the scale of the entire brain, some of the key neural mechanisms underlying sleep are implemented and regulated at the level of local cortical circuits[24,25]. Consequently, when the pressure for sleep increases, an awake brain can start displaying local sleep-like patterns of activity such as delta or theta waves[21,22,29,42]. These bouts of local sleep-like activity, in the form of large-amplitude slow waves, are both time-dependent (i.e. increase with time spent awake) and use-dependent (i.e. depend on the level of activation of a given brain region)[29,30]. The occurrence of slow waves has been linked to perturbations of information processing and task-related errors or attentional lapses in animal and human intracranial data[21,22]. A similar relationship between local sleep and behavioural errors has been evidenced when detecting slow waves in non-invasive scalp recordings of sleep-deprived[29,30] or well-rested[31] individuals.

Importantly, our study proposes a link between sleep-like slow waves in normal wakefulness and both behaviour and subjective experience. Specifically, we showed that, within the same individual, slow waves temporally precede different types of attentional lapses. At the behavioural level, we observed that sluggish responses (slow responses and misses) were associated with an increase in slow waves over posterior electrodes (Fig. 5a, c). Conversely, impulsive responses (hasty responses and FAs) were associated with an increase in slow waves over frontal electrodes (Fig. 5a, b). At the phenomenological level, compared to task-focused (ON) reports, MW was preceded by an increase in slow-wave density, amplitude, upward and downward slopes over frontal electrodes (Fig. 4a). Compared to ON reports, MB was also preceded by an increase in slow-wave density over frontal electrodes, but we additionally observed an increase in slow-wave upward and downward slopes over posterior electrodes (Fig. 4b). A direct comparison between MW and MB reports shows that MB reports were associated with slow waves with steeper upward slopes over posterior electrodes and smaller amplitude over frontal electrodes (Fig. 4c). Finally, focusing on shorter time windows (Supplementary Figs. 4–6), we showed that slow waves best predict mental states (MW or MB vs. ON) in the last few seconds before a probe, in line with previous findings[43,44].

These results imply a spatial and temporal relationship between markers of local sleep and behavioural errors: local sleep events, as measured through the presence of slow waves, occurring at the right time (i.e. during stimulus presentation) and in the right place (i.e. in the brain regions involved in the task) are predictive of distinct behavioural and phenomenological aspects of attentional lapses[11].

Importantly, our results are largely consistent with previous findings on the neural correlates of attentional lapses. Most of these studies focused on MW, although often defined as any mental state that is not on task (i.e. attentional lapses, which include MW and MB as defined here). Early functional magnetic resonance imaging (fMRI) studies showed that MW in this sense was associated with the activation of the Default Mode Network (DMN)[45,46]. Relevant to our local sleep interpretation, the DMN is also suggested to be involved in dream generation during sleep, consistent with the idea that the DMN supports a broad array of experiences that are decoupled from the environment[47]. Furthermore, lesions within the DMN are associated with a decrease in both MW[48] and mind dreaming[49]. However, recent findings suggest a complex relationship between the DMN and sponta-neous experiences. Unlike what was initially hypothesized, the DMN is now considered to be involved in both task-unrelated and task-related processes[43,50], depending, e.g., on environmental demands or the vividness of individuals' experiences[51,52]. Based on our results, we speculate that local sleep plays a key role in the complex relationship between the DMN and spontaneous experiences. Previous studies have indeed shown that a state of low alertness could induce the phasic activation of the DMN[53]. We speculate that slow waves occurring within the DMN could lead to episodes of MW by disconnecting individuals from their environment[54]. Indeed, slow waves during sleep have been pro-posed to be responsible for disconnecting sleepers from their environment[27,55], as they are accompanied by a phenomenon of neuronal silencing that can disrupt the processing of external inputs. If slow waves trigger specific occurrences of MW, other mechanisms could be responsible for the stability of these episodes[56]. Further investigations, including source localization or simultaneous recording of EEG and fMRI[57,58], promise a deeper understanding of the mechanisms underlying these attentional lapses.

Our work might also help reconcile previous findings on the EEG correlates of MW. A seminal study reported a reduction of alpha and/or beta oscillations during MW[59], whereas others have reported an increase[60–62]. However, the relation between alpha and sleep is complex: alpha power is low both when participants are fully alert and, on the contrary, when approaching sleep onset[63,64]. Perhaps, the divergent results obtained in previous studies regarding MW and alpha oscillations could be explained by different baseline levels of alpha oscillations in these studies. Likewise, past pupillometry results on attentional lapses have been similarly inconclusive. While most studies found a dampening of stimulus-locked increases in pupil size during MW (e.g. refs. [46,65,66]), results diverge for baseline pupil size, which has been reported as either increasing (e.g. refs. [67,68]) or decreasing (e.g. refs. [46,69,70]). When distinguished from MW, MB has also been associated with reduced pupil size compared to task-focused states[10,65]. Our results largely align with the latter results, with both MW and MB being characterized by a decrease in pre-probe pupil size (Fig. 2e), which goes together with low vigilance ratings (Fig. 2d) and an increase in slow-wave occurrence (Figs. 3 and 4). In addition, the complex relations between pupil size and MW may be partly explained by the fact that pupil size does not index only arousal but also correlates with motivation[71] and cognitive load[72], which also correlate with MW[2]. In contrast, the sleep-like nature of wake slow waves would make them an unambiguous marker of sleepiness. Furthermore, slow waves are detected at the electrode level and can therefore indicate how brain regions respond differently to sleep pressure.

Other than MW and MB, we foresee that the spatio-temporal properties of slow waves might predict other types of spontaneous experiences. For example, previous research focusing on the hypnagogic period at sleep onset has shown that slow-wave-like activities are predictive of the occurrence of spontaneous imagery[73], the intensity of thoughts[57] and can discriminate between different contents of spontaneous experiences[74]. Like-wise, during sleep, it has been reported that local modulation of slow-wave power is predictive of the occurrence and contents of dreams[75]. Local sleep, therefore, might be related to the type and content of spontaneous experiences not only during wakeful states, as we showed here, but also during sleep–wake transitions and sleep.

Our results also speak to the broader question of how different brain regions participate in shaping the stream of consciousness. Slow waves, considered as a perturbation of the normal wake activity of local cortical networks, could reflect the functions of

these networks under normal situations. For example, we observed that slow waves in frontal regions were associated with FAs (Fig. 5b), which aligns well with the role of frontal cortices in executive functions and response inhibition[76]. Conversely, slow waves in the back of the brain were associated with misses (Fig. 5c), which is consistent with the involvement of parietal cortices in sensorimotor integration[77]. At the phenomenological level, frontal slow waves were associated with MW, whereas posterior slow waves were associated with MB. Thus, our results could speak to the debate on the respective involvement of frontal and posterior cortices in supporting different conscious states[78,79]. Our results suggest that a perturbation of frontal cortices leads to unconstrained thoughts (MW) rather than the loss of awareness, but that awareness decreases when posterior regions momentarily go offline, a pattern similar to the neural correlates of dreaming during sleep or spontaneous thoughts during wakeful rest[57,75]. However, frontal and posterior slow waves do not only differ by their location but also in terms of spatial expanse: slow waves in frontal electrodes appear more focal, whereas slow waves in posterior electrode are more widespread (Supplementary Fig. 10). Thus, the loss of awareness during MB could reflect a tendency for posterior slow waves to involve a broader fronto-parietal network. This is in line with theories attributing an essential role of fronto-parietal connections in the emergence of consciousness[80–82].

We speculate that the slow waves we report here are generated by similar neural mechanisms as slow waves in sleep. Our speculation rests on the profile of slow waves characterized in (i) time (Fig. 3a) and (ii) space (Fig. 3b, c), and the relationships between slow-wave properties and (iii) time spent on the task (Supplementary Fig. 3), (iv) subjective vigilance and (v) pupil size. While these five lines of evidence are correlational, we note that together, they imply a high degree of similarity between slow waves in waking and sleep. This, in turn, allows us to interpret wake slow waves as local sleep. Importantly, as our study builds on convergent lines of indirect evidence to argue that the slow waves we measured in waking participants are sleep-related, this interpretation is tentative and we do not wish to suggest that the presence of slow-frequency oscillations in wake EEG is by itself sufficient to establish a relation to sleep. Here, for example, we paired these observations with objective (pupil size) and subjective (vigilance ratings) markers of fatigue and checked that slow waves increased with the time spent on task. Future studies could use direct evidence from intracranial recordings or sleep deprivation to more solidly establish this interpretation. Intracranial recordings in particular can confirm whether wake slow waves are accompanied by episodes of neuronal silencing, as for sleep slow waves, and would help the identification of the exact neural mechanisms that generate the slow waves we report here.

As our task involved the continuous presentation of visual stimuli, we could not fully disentangle the occurrence of slow waves and task-related events. Yet, we showed that slow waves differ from typical responses evoked by stimuli or participants' responses (Supplementary Fig. 11). Future studies could include task-free resting-state sessions to compare the occurrence and properties of slow waves occurring during a task and without a task. Similarly, further investigation could help determine to which extent our findings generalize to everyday situations (e.g. driving, attending a lecture, reading, etc).

It is worth noting that the tasks used here (Sustained Attention to Response Task (SARTs)) are rather undemanding and could favour sleepiness and local sleep compared to more difficult and engaging experimental paradigms. Generalizing our findings to different experimental contexts or more naturalistic settings would need further experimental validation[11]. From our findings,

we predict that slow waves would predict the occurrence of attentional lapses and MW in situations where participants feel rather sleepy and to a lesser extent or not at all in situations where participants are well awake, motivated and highly engaged. We also speculate that slow waves are not the only mechanism that underlies sensory decoupling in wakefulness and sleep[55]. Other factors, such as the neuromodulation of brain activity, may be critical in clarifying the total set of mechanisms underlying MW[9,83] and MB, and more broadly attentional lapses.

In conclusion, we show here that attentional lapses occurring in the context of an undemanding task are accompanied by sleep-like slow waves, even when participants are well-rested. Furthermore, the location of slow waves is predictive of certain behavioural and phenomenological properties of these lapses. Thus, we propose that these slow waves reflect local intrusions of sleep within waking and constitute a mechanistic and proximate cause to explain attentional lapses. Identifying a proximate mechanism of attentional lapses could inspire novel applications leveraging brain-machine interfaces in educational or professional settings.

## Methods

**Participants**. Thirty-two ($N = 32$) healthy adults were recruited and participated in this study. Six individuals were not included in our analyses because of technical issues during recordings or an abnormal quality of physiological recordings assessed through a post hoc visual inspection of the data. The remaining 26 participants (age: $29.8 \pm 4.1$ years, mean ± standard deviation; ten females) were included in all analyses except for one individual for whom we do not have eye-tracking data. Participants provided written informed consent prior to participating in the study. The protocol was approved by the Monash University Human Research Ethics Committee (Project ID: 10994).

**Experimental design and stimuli**. Participants were seated in a dimly lit room with their chin resting on a support at ~60 cm from a computer screen. All task instructions and stimuli were displayed and button responses were collected via the Psychtoolbox extension[84] for Matlab (Mathworks, Natick, MA, USA).

The experimental design consisted of two modified SARTs[85] in which participants were instructed to pay attention to a series of pictures of human faces in the Face SART blocks or digits in the Digit SART blocks. The order of Face or Digit blocks was pseudo-randomized for each participant. Each block lasted ~12–15 min. Participants were allowed to rest between blocks. Participants performed three Face SART blocks and three Digit SART blocks for a total duration of $103 \pm 19.7$ min (mean ± standard deviation) from beginning to end. Each type of the Face and Digit SART was preceded by a brief training session (27 trials) on each SART. During this SART training session, feedback on the proportion of correct trials and average RTs was provided to participants. Participants were encouraged to prioritize accuracy over speed.

Face and digit stimuli were presented continuously, each stimulus appearing for a duration of 750–1250 ms (random uniform jitter). Face stimuli were extracted from the Radboud Face Database[86] and consisted of eight faces (four females) with a neutral facial expression and one smiling female face. Digits from 1 to 9 were displayed with a fixed font size. For the Face SART, participants were instructed to press a button for all neutral faces (Go trials) but to avoid pressing the response button for the smiling face (NoGo trials). The order of faces was pseudo-randomized throughout the entire task (i.e. we permuted the presentation order every nine stimuli and we did not present twice the same stimuli in a row). For the Digit SART, participants were instructed to press a button for all digits except the digit 3 (NoGo trials), with the order of the digits pseudo-randomized as well.

During the SART, we stopped the presentation of stimuli at random times (every 30 to 70 s, random uniform jitter) with a sound and the word "STOP" displayed on the screen. These interruptions allowed us to probe the mental state of the participants with a series of eight questions (including one conditional question; see Supplementary Methods). In particular, we instructed participants to report their attentional focus "just before the interruption". Participants had to select one of the four following options: (1) "task-focused" (i.e. focusing on the task, ON), (2) "off-task" (i.e. focusing on something other than the task, which we define here as MW), (3) "MB" (i.e. focusing on nothing), (4) "don't remember". As the fourth option accounted for only 1.1% of all probes (i.e. <1 probe per participant on average) and since previous studies do not always distinguish between these options (e.g. ref. [87]), we collapsed the third and fourth options as MB in all analyses. We also instructed participants to rate their level of vigilance, reflecting "over the past few trials", with a 4-point scale (Fig. 2d; from 1 = Extremely Sleepy to 4 = Extremely Alert). Each of the 12–15 min SART blocks included ten interruptions (in total, 30 interruptions for each SART task and 60 interruptions

per participant). Participants were informed of the presence of interruptions and the nature of each question before starting the experiment. The mental state categories (ON, MW and MB) were also explained to participants orally and in writing.

**Physiological recordings and preprocessing**. High-density scalp EEG was recorded using an EasyCap (63 active electrodes) connected to a BrainAmp system (Brain Products GmbH). A ground electrode was placed frontally (Fpz in the 10–20 system). Electrodes were referenced online to a frontal electrode (AFz). Additional electrodes were placed above and below the left and right canthi, respectively, to record ocular movements (electrooculogram, EOG). Two electrodes were placed over the deltoid muscles to record electrocardiographic (ECG) activity. EEG, EOG and ECG were sampled at 500 Hz. Eye movements and pupil size on one eye were recorded with an EyeLink 1000 system (SR Research) with a sampling frequency of 1000 Hz. The eye tracker was calibrated at the start of each recording using the EyeLink acquisition software.

The EEG signal was analysed in Matlab with a combination of the SPM12, EEGlab[88] and Fieldtrip[89] toolboxes. The raw EEG signal was first high-pass filtered >0.1 Hz using a two-pass fifth-order Butterworth filter. A notch filter was then applied (stopband: [45, 55] Hz, fourth-order Butterworth filter) to remove line noise. Electrodes that were visually identified as noisy throughout the recording were removed and interpolated using neighbouring electrodes. Finally, the continuous EEG data were segmented according to probe onsets on a 64 s window ([−32, 32] s relative to the probe onset); the average voltage over the entire window (64 s) was then removed for each electrode and probe.

Pupil size was analysed with custom functions in Matlab and corrected for the occurrence of blinks (see ref. [34] and Supplementary Methods). Pupil size was averaged over the stimulus presentation window for each trial (window length: 0.75–1.25 s). Pupil size values in Fig. 2e were computed by averaging the pupil size in all trials within 20 s preceding the probe onset and then by discretizing them into 5 bins across all probes for each participant and task, to normalize pupil size across participants[34].

Raw behavioural, eye-tracking and EEG data are publicly available[90] as well as the codes used to preprocess and analyse these data[91].

**Behavioural analyses**. Go trials were considered incorrect (Miss) if no response was recorded between stimulus onset and the next stimulus onset. Conversely, NoGo trials were considered incorrect (FA) if a response was recorded between stimulus onset and the next stimulus onset. RTs were computed from the onset of the stimulus presentation. Trials with RTs shorter than 300 ms were excluded from all analyses (so not considered correct nor incorrect). In all subsequent analyses, we focused only on trials within 20 s from probe onset. The choice of a 20 s window allowed the inclusion of two NoGo trials for each probe, while focusing on trials that are relatively close to the probe onset and subsequent subjective reports. Since previous studies sometimes examined shorter pre-probe time windows (e.g. ref. [43]), we also analysed the data with shorter time windows (Supplementary Table 3 and Supplementary Figs. 4–6).

**Detection of slow waves**. The detection of sleep-like slow waves in waking was based on previous algorithms devised to automatically detect slow waves during NREM sleep[30,36]. First, the preprocessed EEG signal was re-referenced to the average of the left and right mastoid electrodes (TP9 and TP10) to match the established guidelines for sleep recordings[92]. Then, the signal was down-sampled to 128 Hz and band-pass filtered within the delta band. A type-2 Chebyshev filter was used to reach an attenuation of at least 25 dB in the stopband ([0.1, 15] Hz) but <3 dB in the passband ([1, 10] Hz). All waves were detected by locating the negative peaks in the filtered signal. For each wave, the following parameters were extracted: start and end point (defined as zero-crossing, respectively, prior to the negative peak of the wave and following its positive peak), negative peak amplitude and position in time, positive peak amplitude and position in time, peak-to-peak amplitude, downward (from start to negative peak) and upward (from negative to positive peak) slopes.

Slow waves in sleep typically have a larger negative peak compared to their positive peak (Fig. 3a) and are predominantly observed over fronto-central channels[36,37] (Fig. 3b). This contrasts with artefacts in the EEG signal caused by blinks, which typically have a large positive component and are more frontally distributed. To reduce the false detection of these artefacts as candidate slow waves, we excluded waves with a positive peak >75 µV. We also excluded waves within 1 s of large-amplitude events (>150 µV of absolute amplitude). Finally, we discarded all waves that were shorter than 143 ms in duration (corresponding to a frequency >7 Hz). We then selected the waves with the highest absolute peak-to-peak amplitude (top 10% computed for each electrode independently) as local sleep slow waves. This 10% threshold was selected based on the visual examination of the distribution of the amplitudes at the subject level (see Supplementary Fig. 12 for the corresponding topography). The mean ± SEM of the threshold voltage across subjects was 30.7 ± 1.5 µV. Finally, we compared slow waves with neural activity related to task events (stimulus onsets and motor responses). To do so, we computed the event-related potentials (ERPs) by averaging, across participants, the

EEG signal band passed around 0.1 and 30 Hz and referenced to the mastoids. These ERPs were locked either on slow-wave start, stimulus onset (for face and digit stimuli separately) or motor responses. The corresponding temporal profiles are shown in Supplementary Fig. 11a–c for electrode Cz.

**Hierarchical drift-diffusion modelling**. Hierarchical Bayesian Drift-Diffusion Modelling (HDDM) was used to extend our analysis beyond simple behavioural metrics and examine the impact of mental states (Supplementary Fig. 9) and slow waves (Fig. 6) on the sub-processes of decision-making. The DDM is a sequential-sampling model of 2AFC decision-making that can be considered an extension of signal detection theory into the time-domain, accounting for full reaction-time distributions as well as choice behaviour[38]. The HDDM package[39] in Python 2.8 was used to fit the drift-diffusion model to the SART data. DDM parameters were estimated using a hierarchical Bayesian method that uses Markov-chain Monte Carlo (MCMC) sampling to generate full posterior distributions of model parameters. The following DDM parameters were estimated: the drift rate for Go trials ($v_{Go}$), the drift rate for NoGo trials ($v_{NoGo}$), the decision threshold ($a$), the decision bias ($z$) and the NDT ($t$). Drift bias ($v_{Bias}$) was computed by taking the difference between the absolute values of $v_{Go}$ (positive) and $v_{NoGo}$ (negative), where greater values indicate stronger $v_{Go}$ drift bias (Supplementary Fig. 7). To examine whether the model could reproduce key patterns in the SART data, posterior predictive checks were undertaken by simulating 100 datasets from the joint posteriors of model parameters and comparing these to the observed data[93] (Supplementary Fig. 8).

To estimate HDDM parameters 8000 samples from the posterior were generated with MCMC methods and the initial 2000 were discarded as burn-in to reduce autocorrelation. HDDM models were first to fit on the trials preceding subjective reports to examine the influence of mental states on decision parameters (Supplementary Fig. 9). HDDM parameters were free to vary by state and task. From the estimated models, we extracted the subject-level point estimates of parameters as the mean of each individual's posterior distribution for a given task (Face and Digit) and mental state (ON, MW or MB). We then fitted HDDM models so as to examine the influence of slow waves (event present vs. absent; Fig. 6). To do so, we considered each EEG electrode separately. For a given electrode, a trial was flagged as being associated with slow waves if the onset of a slow wave was detected for this electrode during stimulus presentation (i.e. between stimulus onset and offset). Parameters were also free to vary by task (Digit vs. Face). When examining the impact of mental states or slow waves on HDDM parameters, we included trials within 20 s of a probe onset (allowing to include two NoGo trials for each probe). From the estimated models, we extracted the subject-level point estimates of parameters as the mean of each individual's posterior distribution for a given task (Face and Digit) and slow waves (present or absent). Statistical comparisons were performed on the subject-level point estimates.

**Statistical analyses**. Statistics were performed using linear mixed-effects modelling (LMEs). In all models, subject identity was coded as a categorical random effect. The task type (Digit or Face SART) was used as a categorical fixed effect in all analyses. Several fixed effects were independently tested in our different analyses: mental state (categorical variable: ON, MW and MB; Figs. 2–4) or slow waves (binary variable: present/absent, Fig. 5). LMEs were run to predict different variables of interest: behavioural variables (misses, FAs, RTs) or physiological variables (pupil size, presence or properties of slow waves). We also used LMEs to estimate the effect of mental states or slow waves on the point estimates derived from the HDDM models (Supplementary Figs. 9 and 6). Model comparisons were performed using a likelihood ratio test to estimate the influence of multi-level categorical variables such as mental states. In practice, we compared a model including mental state (along with possibly other random and fixed effects) with a model not including mental state as a predictor. All models and model comparisons are described in Supplementary Table 1. In the "Results" section, we report the likelihood ratio test as $\chi^2$(d.f.), where $\chi^2$ is the likelihood ratio test statistic and d.f. is the degree of freedom[94,95]. When several model comparisons were performed for the same analysis using the likelihood ratio test, a Bonferroni correction was applied to the statistical threshold. To indicate the magnitude and direction of the effects, we report the estimates ($\beta$) and CI for the contrasts of interest (MW vs ON, MB vs. ON, MB vs. MW). All models performed are described in Supplementary Table 1. For topographical maps, clusters were identified using a cluster-permutation approach. In practice, for each electrode, we extracted the $t$ values and $p$ values for the effects of interest. Clusters of neighbouring electrodes were defined as electrodes with $p$ values below 0.025 (cluster alpha). Once the clusters were defined, a comparison was performed with permuted datasets and we used a Monte Carlo $p$ value threshold of 0.05 to identify the significant clusters ($p_{cluster}$, see Supplementary Methods for details). This threshold was corrected for multiple comparisons when several non-independent cluster permutations were performed (e.g. Figs. 4–6 and Supplementary Figs. S2–4) in order to keep the type-1 error rate constant across analyses. Raincloud plots (Figs. 1 and 2 and Supplementary Fig. S9) were obtained with the RainCloudPlots toolbox[94].

**Reporting summary**. Further information on research design is available in the Nature Research Reporting Summary linked to this article.

## Data availability

The source data are also publicly available at https://osf.io/ey3ca/?view_only=680c39e7065649c3b783a4efec0a1a94[90]. Source data are provided with this paper.

## Code availability

All code generated for this study's analyses are publicly available at https://github.com/andrillon/wanderIM[91].

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

## Acknowledgements

T.A. and N.T. were supported by Australian Research Council Discovery Projects (DP180104128 and DP180100396) and National Health Medical Research Council Ideas Projects (APP1183280). T.A. was supported by a Long-Term Fellowship from the Human Frontier Science Programme (LT000362/2018-L). J.W. was supported by Australian Research Council Discovery Early Career Researcher Awards (DE170101254). This work was supported by JSPS Grant-in-Aid for Transformative Research Areas (B) Grant Number 20B101. We thank Devangna Tangri for her help in data collection and Giulio Bernardi for sharing his slow-wave detection algorithm.

## Author contributions

Design: T.A., T.M., J.W. and N.T. Data collection: T.A. and T.M. Analyses: T.A. and A.B. Manuscript: T.A., A.B., J.W. and N.T.

## Competing interests

The authors declare no competing interests.
