## [Peer Review File · Nature Communications]

REVIEWER COMMENTS

Reviewer #1 (Remarks to the Author):

Andrillon NCOMMS-269095

I apologise for the tremendous delay in reviewing this manuscript, it has been more work and urgent tasks that I expect in the beginning of term and it was difficult to concentrate in scientific tasks when everything is management in covid year I 2020.

The authors use go/nogo tasks and experience sampling to capture the mindwandering and mindblinking events when participants spend long time on boring tasks. This is a well performed study and it brings a fresh view with clear hypotheses and a robust analysis to a pervasive event of the mind, attentional lapses. This is understudied and commonly assume that are rare and only important in the context of shift work, driving, or sustained attention demanding tasks (operating machinery, invigilating industrial processes (like Chernobyl)). More important is that the authors define well the neuroscience markers they think will help explain in part the lapses and its differences, a fleeting moment of sleep like brain event called local sleep.

A few comments and critiques that I hope can improve this fine paper in interpretability, robustness and readability. I hope the authors are willing to engage with me in scientific interaction. My key issue for interpretation is that except in the first analysis of the behaviour, there are no other direct comparisons between MW and MB to address the direct differences between them. It is important to define this as the paper claims clearly that separates aspect of the lapses but the evidence in somehow presented in an indirect manner in the details of the paper. I think the manuscript is elegant in the analyses and description of the results so I described here my interpretation and critiques with the aim of improving this piece of work.

Out of curiosity I wondered if the authors tried to normalize the MW and MB results in errors and RTs by each participant on-task values (either average or previous On task trial). This could reveal interesting patterns about the relationship between the external vs internal attentional focus that may inform the discussion. I specifically mention this since the variability in SART tasks and in drowsiness between people is big and this could reveal which participants carry the effect and whether they have a particular relationship between local sleep and content more than the others.

For the first take on vigilance I celebrate the use of both the subjective ratings of the participant and the pupil size, and also even if correlated (should that be shown explicitly), it is interesting to only find reliable difference for subjective ratings between MW and MB but not in pupil measurements. How is that related to content and local sleep?

Also I imagine that the authors have tried to calculate complexity measures (Lempel-ziv, etc) as they had done in previous work, maybe this is left for another paper or maybe it was already too much to calculate a neural measure of transition to sleep since the focus was local sleep itself. Interested to hear your thoughts on the matter. We have in the past also use Dimension of activation with hypnagogic work with positive results.

IT was great to see the characterization of the local sleep waves and the differences between MB and ON, and between MW and ON. But there was no reliable evidence for differences between MB and MW, right? If I read the results correctly, despite the visual differences between MB and MW in density and slope, the statistical analyses does not show reliable differences. This is a sad outcome for the hypotheses of neural diff between types of lapses. If this is the case and my interpretation is correct this should be highlighted as a result and mentioned and discussed afterwards, for the sake of the hypothesis.

This pattern of reporting is repeated in page 13 lines 244-255, where the association between false alarms and misses, and in particular the association of local sleep amp to MB and MW 254-255 "local sleep amplitude and slope was larger for MW compared to ON reports in frontal electrodes only whereas local sleep amplitude was larger for MB compared to ON reports in posterior channels only (pcluster<0.05, Bonferroni corrected cluster threshold; Figure 4b)."

So, again, no reporting of lack of diff or not enough evidence for diff between MB and MW. To be able to claim strongly the differences between the neural markers of local sleep preceding the lapses a direct statistical comparison between MB and MW would show directly the claims that are described later in the discussion and in the abstract. "Furthermore, the location of local sleep distinguished sluggish versus impulsive behaviors, mind wandering versus mind blanking." I think it is important to differentiate a clearly difference to the focus of the task between the types of lapses (MB vs ON and MW vs ON) to a true differentiation between lapses (MB vs MW) in local sleep markers and its associations. Please comment and correct me if I am wrong in my interpretation of the results.

I think it is great to do the modelling in this case as it helps differentiate between go and no go responses in a principled manner in terms of processes happening and possible underlying mechanisms. Yet again, the results are not directly linked to the type of lapse (MB or MW) but to the electrode space (in evidence accumulation I mean) and behavioural parameters. So this is a cool result in its on right and happily complementary to a paper we have under review somewhere else (<https://www.biorxiv.org/content/10.1101/2020.07.23.218727v1> no need to cite, just flagging cause there are no other papers on DDHM and arousal), but does not speak directly to the lapse types unless you link it.

I will wait for the response to these comments to decide to further engage in details of interpretation and sleep and consciousness frameworks. But I highlight a few papers that maybe will be useful for the discussion about content of the mind in lapses of attention related to sleep onset or fatigue.

Other interesting paper related to content in Sleep onset is "EEG Power Associated with Early Sleep Onset Images Differing in Sensory" where aspect of local sleep could be derived from its fine reading

Content http://dreamscience.ca/en/documents/publications/_2001_Nielsen_Germain_Reprint_SRO_4_83-90_EEG_power_at_Sleep_Onset.pdf

Such as changes in local theta and delta associated with specific content.

A study of interest that may be informative is the single case repeated mind wandering in the transition to sleep that we published with Prof Valdas Noreika (Noreika et al 2015). In that paper we show how more the participant trained himself in come out of the transition when the hypnagogic content was interrupted by a dream like bizarreness event or object. " In particular, an increase in the frequency of Hori stages 4 and 5, and a decrease in the frequency of Hori stage 2 were observed in the last 20 s before a button press (report of hypnagogic event), suggesting that the occurrence of unpredictable hypnagogic experiences may depend on a rapid increase of drowsiness." The key issue here is that hori 4 is defined by flattening of the EEG and hori 5 by theta ripples, also called sleep-like slow waves of N1 (Hori refs

Hori, T., Hayashi, M., and Morikawa, T. (1994). "Topographical EEG changes and hypnagogic experience," in *Sleep Onset: Normal and Abnormal Processes*, eds R. D. Ogilvie and J. R. Harsh (Washington, DC: American Psychological Association), 237–253. doi: 10.1037/10166-014

Hayashi, M., Katoh, K., and Hori, T. (1999). Hypnagogic imagery and EEG activity. *Percept. Mot. Skills* 88, 676–678. doi: 10.2466/pms.1999.88.2.676

The other result from the 2015 paper that it is useful for this manuscript is the change in theta (slow wave-like) power before the phenomenological report, that differed between types of hypnagogic experiences (linguistic vs visual/contextual).

"Linguistic intrusions were associated with the lower theta power than perceptual imagery reports in the 60–40 s [$t(54) = 3.43, p < 0.005, d = 1.04$] and 40–20 s [$t(54) = 5.7, p < 0.0005, d = 1.7$] time windows (see Figure 3A). Furthermore, theta power increased for the linguistic intrusions in the last 20 s compared to 60–40 s [$t(43) = 2.47, p < 0.05, d = 0.38$] and 40–20 s [$t(43) = 3.07, p < 0.005, d = 0.45$] time windows. Contrary to this, theta power decreased for the perceptual imagery reports in the last 20 s compared to the 40–20 s time window [$t(11) = 3.33, p < 0.05, d = 0.96$]."

Tristan bekinschtein

Reviewer #2 (Remarks to the Author):

Review of Andrillon and colleagues.

This paper presents an interesting and important study that examines the relationship between EEG patterns during a demanding task context (the SART) and the occurrence of different mental states (On Task, Mind-wandering and Mind-blanking). Using an algorithm that detected high amplitude delta waves in the EEG spectrum, the authors found that these predicted both behaviour and subjective reports of different states. This suggests that high amplitude gamma activity often seen during sleep is an important marker for states of lower vigilance.

I think this is a well-motivated study, that is generally well conducted and that has very important implications for understanding patterns of ongoing thought. There are a few places in which the authors need a more updated account of contemporary accounts of ongoing thought.

However, the biggest issue with the paper as it stands is that its argument is somewhat circular and this framing is unfortunately likely to confuse future research moving forward. The notion that the pattern identified by the algorithm is detecting local sleep, and consequently that all experiences linked to this pattern are therefore an example of 'local sleep' is a reverse inference, and one that the authors should not make the centre of their argument. To play devil's advocate the data in this paper could also be used to question the specificity of the algorithm for detecting sleep per se given that it also predicts other experiential states. In process terms, this pattern may simply reflect the role that decoupling when we sleep or when we shift focus away from the task (Smallwood, 2013, *Psych Bulletin*). I am not strongly against the argument that the authors propose because the SART is a really boring task, but the reality is that there are lots of different ways that these data could be explained rather than by assuming that because a pattern we see when we sleep predicts a particular state then that state must involve falling asleep.

Instead, why not report what was found: large amplitude delta waves are important predictors of an individual's experiential state when it is not focused on a demanding external task. This is an entirely unambiguous description of the results and it completely removes the circularity that is present in the current manuscript. It may well be that local sleep is linked to some features of ongoing cognition, and I think this is a great question that the authors could examine in subsequent studies, but to my mind nothing in this current paper supports the use of language of detecting local sleep etc... that occurs throughout the paper. Instead it would be easy to use an argument that a marker that is present in sleep was applied to vigilance data and found to make meaningful predictions of attentional states, thus removing the notion of local sleep from the title and the bulk of the paper and minimise reverse inference. They can then discuss why this marker predicts attentional states in the discussion. I would like to make it clear that in my opinion that this aspect of the language does not take away from the fact that (i) the study is well conducted, (ii) the results are important. I actually think that changing the authors' language is likely to make this study more important in the future because it opens up the possibility that they have discovered that high amplitude delta waves (sleep related or not) may become a powerful way to understand certain mental states.

20 second window for evaluating the links between brain and behaviour. The authors use a 20 second window to examine experiential states, however, while this has been used in the past (i.e. Christoff et al., 2009) this is in fact a rather long window and they should justify why they used such a long one to predict experience when EEG can resolve things much faster than fMRI can. In work from my lab we have employed 6 second windows successfully (Turnbull et al., 2019, *Nat Comms*; Sormaz et al., 2018). We do this because shorter windows have the advantage of being

less ambiguously linked to subsequent experiential reports rather than things that systematically precede it (like local sleep for example). This is especially important because we know that changes over this length of time influence the likelihood of reporting off task states (e.g. Turnbull et al., 2018, Neuroimage) and that over a similar time frame changes in neural activity also occurs and that this is predictive of reports of off task thought (Turnbull et al., 2020, Scientific Reports). To address this it may be possible for the authors to use a shorter window, or based on the evidence I just described, evaluate whether changes over this period matter for the reports that participants subsequently give.

Discussion of the neural correlates of off-task thought is too simplistic. Although initial views argued that the DMN supports off task thought, we now know that this is an oversimplification and the authors should try to reflect this more nuanced view in the discussion. First, using multiple experience sampling questions in a data driven manner we know now that being off-task is related to reductions in task positive systems, such as the dorsal attention network (Turnbull et al., 2019). It is true that activity in the DMN can be important in the off-task content, e.g. Konu et al., (2020, Neuroimage) found activity in vmPFC was linked to the expression of off task social content. However, using a similar experience sampling protocol we found using both representational similarity analysis (Sormaz et al., 2018, PNAS) and standard whole brain univariate analysis (Turnbull et al., 2018, Neuroimage, 2019, Nat Comms) that the DMN is important when people report experiences with task focused detail in a working memory task. This effect is likely to be targeted on the posterior cingulate cortex (Turnbull et al., 2019) and we suspect it may serve this function even in tasks that are default mode network friendly, such as self-reference (Murphy et al., 2019, PLOS One). Complicating things further regions like the dLPFC can also play a role in off task states (Turnbull et al., 2019) perhaps playing a role in the prioritisation of information relevant to the individuals goal state. It would be useful to try to incorporate the complexity that these studies raise in a revision, perhaps commenting on how their results, and the application of EEG, could help move forward the question of the role of the DMN in cognition. For example, it struck me when reading this paper whether when there are high amplitude delta waves dominate the default mode network becomes decoupled from the task which could explain why it can play a role in both task relevant and off task states.

Signed

Jonathan Smallwood

Minor issues

It might make sense in the discussion to explain that the use of the SART as a paradigm may have contributed to some of the patterns the authors observe. This is because the SART requires participants maintain attention for prolonged amount of time with minimal external support and we know that the neural correlates of both on tasks and off task states change when tasks are designed so that they have minimal vigilance load (e.g. the 0 – back / 1 back paradigms used by Sormaz et al., etc...). The use of multiple tasks conditions in studies like this is helpful because it provides clarity on the contextual conditions within which we might expect the results to emerge (Smallwood and Andrews-Hanna, 2013). Since this was not done in this study it should be mentioned. This is not a limitation because power is lost when the experiment is divided into multiple conditions but it is nonetheless important to mention it because we don't know the contextual specificity of the effect yet.

Page 7. "Taken together, these results suggest that MW and MB can decrease performance through different ways: MW facilitates impulsivity, evidenced by faster RT and more false alarms, whereas MB facilitates sluggishness, evidenced by slower RT and more misses."

Unless I misunderstood the analysis, could this also indicate that MB is fully disengaged when MW was only partially disengaged? (i.e. it was not faster than on task and MB was worse than MW?).

Page 8 and the Discussion on pupil dilation. The author may be interested to note that Konishi and colleagues (2017, Cognition) also found that on tasks states had heightened pupils relative to off task states but only those with a temporal focus on the past, or where highly intrusive in nature.

Reviewer #3 (Remarks to the Author):

In this study Andrillon and colleagues investigated the relationship between local sleep, mental states and performance on a sustained attention to response task (SART) in 26 healthy awake individuals. They report that the occurrence of large EEG slow waves ('local sleep') is not only associated, in a region specific manner, with errors in the task (hits vs. misses), but also with distinct mental states (mind blanking vs. wandering).

The existence of local sleep in wakefulness and its regionally specific effects on behavior have been reported previously in both humans and rodents. What is new and noteworthy here is the finding that large EEG slow waves during wakefulness are associated with reports of mind blanking/wandering in a regionally specific manner. If confirmed, this finding would be of interest for (neuro)science in general, beyond the field of sleep.

My main concern with this paper is that it lacks the crucial proof that the large EEG slow waves they investigate are truly sleep related. Such a proof would require the demonstration that mindblinking/wandering and the associated EEG slow waves increase as a function of time spent awake (that is, as a function of homeostatic sleep pressure), or intracranial recordings documenting neuronal off-states typical of sleep slow waves, as has been done in previous studies. Here the authors use correlations with subjective vigilance scores and pupil size as a proxy, which in my view is insufficient to sustain such a strong claim. Neither EEG low-frequency oscillations nor pupil size are unambiguous markers of sleepiness/local sleep.

Other comments:

2) The EEG recording used to study the association between slow waves and mind states appears to contain both stimulus-induced and spontaneous activity. This is an issue, as large stimulus-evoked potentials can sometimes be observed in the wake EEG and cannot be considered 'local sleep'. Although the large waveform shown in Figure 1a represents an average of 26 subjects, it appears extremely sharp and is preceded by a fast and clear positive-negative-positive deflection – could this reflect such stimulus-evoked activity?

3) Methodological choices are not always justified and therefore appear arbitrary at times:

- In Figure 3c, the authors show the distribution of slow wave parameters across different mind states. To quantify the effects, they run a "mixed-effect model analysis focusing on local sleep events before each probe and across all electrodes". Why is this "main" effect of mind state (model comparison with likelihood ratio) evaluated on the average across electrodes (considering that LOCAL sleep is the main focus) and the contrasts performed electrode by electrode (shown in Figure 4)?

- Why were different time windows prior to probes (20s vs 30s) used for different analyses (lines 480-81)?

- Why was a threshold of $p < 0.01$ used for the cluster permutation approach, while for the Monte-Carlo simulations, a threshold of $p < 0.05$ was used?

- Why were mind blanking and "don't remember" trials grouped within the same category, considering that "don't remember" trials are few and may have distinct neurophysiological substrates (see for instance Fazekas Sleep Med Reviews 2018 for an analogy with sleep experiences)?

- The authors only selected waves with the highest absolute peak-to-peak amplitude (top 10%). Why did they chose this threshold? How variable was the absolute amplitude of these waves

between subjects? Is the incidence of such local sleep events (3-8/min depending on electrodes) comparable to other studies?

- Why were two different tasks performed (face and digit)?

- Why are only false alarms shown in Figure 2 and not misses (lines 137-140)?

4) It is sometimes difficult to follow the authors' conclusions based on the data that is presented:

- Lines 177-179: "Taken together, these results suggest that MW and MB can decrease performance through different ways: MW facilitates impulsivity, evidenced by faster RT and more false alarms, whereas MB facilitates sluggishness, evidenced by slower RT and more misses." This conclusion is not entirely correct because no difference in false alarms was found between MB and MW.

- Slow wave slope and amplitude are taken as equivalent markers of local sleep (Figure 4b, text lines 252 and following), although their topography does not overlap (ex. MW/ON contrast) and without a clear justification. Previous studies have shown that the parameters likely reflect distinct neurophysiological processes. Also, it is not clear why the authors report on downward but not upward slope and whether they assessed slow wave density in this context. Results for all parameters should be reported.

- Line 200: "Both the temporal profile and topographical distributions of local sleep detected during the tasks (Figure 3a-b) resemble the slow waves observed in NREM sleep". Please provide the topoplots for sleep, not only the waveform, so that one can directly compare.

5) Methodological clarifications:

- Please clarify whether the subjects were well rested when doing the task, or whether the experiment was performed in conditions that maximize the occurrence of local sleep (i.e. sleep restriction, specific time of the day, etc.).

- Please specify each time you use 'slope' whether you mean downward or upward slope and if you report only one, why (no effect for the other)?

- Were stimuli constantly presented or was there an interval between visual stimuli? If yes, how long was it?

- How many probes were collected per subject?

- Was the slow wave detection performed after removing average voltage around entire window (64s), as specified in line 466?

6) Minor comments

- Line 456: Two electrodes were placed over the deltoid muscles to record electrocardiographic (ECG) activity. Do the authors mean electromyographic activity?

- Supplementary table, line for Figure 4b: where does the mind state fit in? Instead of "Local sleep", do the authors mean 'mind state' (as a predictor instead of local sleep)?

- The asterisks in Figure 2 ***, which refer to the likelihood ratio, are confusing, as one may erroneously interpret them as a difference between the ON and MB conditions

- Figure 4: It is not immediately clear (even if there is a line separating the two rows) that the labels "MW vs ON" and "MB vs ON" refer only to the lower line. In addition the title 'local sleep' in Figure 4b is misleading, as this formulation is used to define the presence/absence of local sleep in Figure 4a, but in the lower line, slow waves parameters are shown instead. It would be better to more clearly separate the two panels, to indicate that only contrasts yielding significant differences are shown, and to replace 'local sleep' with 'slow wave parameters'.

- The explanations on the Hierarchical Drift Diffusion Model are very difficult to understand, even after reading the online methods
- Figure 1, please explain how the slow waves were aligned (based on the first zero-crossing?)

REVIEWER COMMENTS

Note: In our responses to Reviewers' comments, we will use text in bold blue font. Portions of Reviewers' comments are in bold font for emphasis.

Reviewer #1 (Remarks to the Author):

Andrillon NCOMMS-269095

I apologise for the tremendous delay in reviewing this manuscript, it has been more work and urgent tasks that I expect in the beginning of term and it was difficult to concentrate in scientific tasks when everything is management in covid year I 2020.

The authors use go/nogo tasks and experience sampling to capture the mindwandering and mindblinking events when participants spend long time on boring tasks. This is a well performed study and it brings a fresh view with clear hypotheses and a robust analysis to a pervasive event of the mind, attentional lapses. This is understudied and commonly assume that are rare and only important in the context of shift work, driving, or sustained attention demanding tasks (operating machinery, invigilating industrial processes (like Chernobyl)). More important is that the authors define well the neuroscience markers they think will help explain in part the lapses and its differences, a fleeting moment of sleep like brain event called local sleep.

Response: **We thank the Reviewer for their positive appraisal of our work.**

A few comments and critiques that I hope can improve this fine paper in interpretability, robustness and readability. I hope the authors are willing to engage with me in scientific interaction. **My key issue for interpretation is that except in the first analysis of the behaviour, there are no other direct comparisons between MW and MB to address the direct differences between them.** It is **important to define this** as the paper claims clearly that separates aspect of the lapses but **the evidence in somehow presented in an indirect manner** in the details of the paper. I think the manuscript is elegant in the analyses and description of the results so I described here my interpretation and critiques with the aim of improving this piece of work.

Response: **The Reviewer is right in pointing out that we should have included a direct comparison between MW and MB in all analyses. As suggested by the Reviewer, such a comparison could bolster our claim that different types of sleep-like slow waves underlie different types of lapses. To address the Reviewer's comment, we now report direct comparisons between MW and MB in all analyses (see below).**

Out of curiosity I wondered **if the authors tried to normalize the MW and MB results in errors and RTs by each participant on-task values** (either average or previous On task trial). This could reveal interesting patterns about the relationship between the external vs internal attentional focus that may inform the discussion. I specifically mention this since the variability in SART tasks and in drowsiness between people is big and this could reveal which participants carry the effect and whether they have a particular relationship between local sleep and content more than the others.

Response: **We thank the Reviewer for this interesting suggestion. Accordingly, for each subject and each task, we normalised the average RTs, % misses and % false alarms obtained for each condition (ON, MW and MB) by the respective average obtained prior to ON probes. Applying the same modelling approach (LMEs and model comparison) as in our initial submission to these normalised data led to very similar results (see Table R1 below). We report this limited impact of normalisation in the Results section (p. 10, l. 199-201) and the new Supplementary Table 2.**

		Normalised	Before Normalisation
Misses (%) in Go trials	State	$\chi^2(2)=33.4$ (***)	$\chi^2(2)=36.0$ (***)
	MW vs ON	-0.0096 [-0.015, -0.0041]	-0.011 [-0.016, -0.0052]
	MB vs ON	-0.023 [-0.032, -0.015]	-0.023 [-0.032, -0.015]
	MB vs MW	-0.014 [-0.022, -0.0055]	-0.013 [-0.021, -0.0047]
False alarms (%) in NoGo trials	State	$\chi^2(2)=88.5$ (***)	$\chi^2(2)=115.9$ (***)
	MW vs ON	-0.24 [-0.29, -0.19]	-0.20 [-0.24, -0.17]
	MB vs ON	-0.20 [-0.26, -0.19]	-0.17 [-0.23, -0.12]
	MB vs MW	0.064 [-0.014, 0.14]	0.028 [-0.028, 0.084]
RT (s) in Go trials	State	$\chi^2(2)=33.4$ (***)	$\chi^2(2)=16.9$ (**)
	MW vs ON	-0.0005 [-0.013, 0.011]	-0.0025 [-0.0096, 0.0045]
	MB vs ON	0.039 [0.022, 0.057]	0.019 [0.0088, 0.030]
	MB vs MW	0.040 [0.022, 0.058]	0.022 [0.011, 0.032]

Table R1 (Supplementary Table 2): Effect of normalisation on behavioural analyses. The analyses of the effect of mental states (model comparisons and post-hoc contrasts) were consistent even after normalizing Misses, False alarms and Reactions Times (RT) for each participant by the average obtained on ON probes ("Normalised" column). The "State" row shows the result of the likelihood ratio test (chi-squared value). Stars denote the corrected significance level (Bonferroni correction for 6 likelihood ratio tests: ***, $p < 0.001$, **, $p < 0.01$, *, $p < 0.05$). The estimates of the post-hoc contrasts (MW vs ON, MB vs ON, MB vs MW) obtained from the mixed effect models are also reported, with the corresponding 95% confidence intervals. Bold fonts signal the significant results ($p < 0.05$ for model comparisons, or confidence intervals excluding 0 for the post-hoc contrasts).

For the first take on vigilance I celebrate the use of both the subjective ratings of the participant and the pupil size, and also **even if correlated (should that be shown explicitly)**, it is interesting to only find **reliable difference for subjective ratings between MW and MB but not in pupil measurements**. How is that related to content and local sleep?

Response: The Reviewer is right in pointing out that there is a significant difference between MW and MB for vigilance ratings (p. 10, l. 211-212) but not for pupil size (p. 11, l. 217-219). In addition, there is a significant within-subject correlation between the subjective ratings and the pupil size as shown in the Figure R1 below:

Figure R1. Distribution of correlation coefficients between pupil size and vigilance ratings for each subject and each task.

Spearman's correlation coefficients were computed between pupil size and vigilance ratings for each subject and each task. Smaller circles and diamonds show the individual coefficients for the Face Task (circles) and the Digit task (diamonds). The mean across participants is also shown (the bigger circle and diamond) and the error bars indicate SEM across subjects. Both distributions were significantly higher than 0, indicating a positive correlation between vigilance ratings and pupil size across the samples.

We now report this correlation in the Results section using mixed-effect models and a model comparison approach, as we did elsewhere in the article (see p. 11 and below). However, the relationship between pupil size and vigilance ratings is not perfect, which could explain the discrepancies between pupil size and vigilance ratings when comparing MW and MB:

“Pupil size did not differ between MW and MB (MB vs MW: $\beta=0.065$, CI: [-0.16, 0.29]), despite the significant correlation between vigilance ratings and pupil size (model comparison: $\chi^2(2)=134.5$, $p<10^{-16}$; see Table S1 for details on the models). This implies that these two measures of vigilance can be differentially sensitive to mental states.”

Also I imagine that the authors have tried to calculate complexity measures (Lempel-ziv, etc) as they had done in previous work, maybe this is left for another paper or maybe it was already too much to calculate a neural measure of transition to sleep since the focus was local sleep itself. Interested to hear your thoughts on the matter. We have in the past also use Dimension of activation with hypnagogic work with positive results.

Response: Computing the Lempel-Ziv complexity before the probes is certainly something we are considering but we have not done this yet. We thank the Reviewer for mentioning his work on “Dimension of activation” as it could indeed represent another interesting direction.

IT was great to see the characterization of the local sleep waves and the differences between MB and ON, and between MW and ON. But **there was no reliable evidence for differences between MB and MW, right?** If I read the results correctly, despite the visual differences between MB and MW in density and slope, the statistical analyses does not show reliable differences. This is a sad outcome for the hypotheses of neural diff between types of lapses. **If this is the case and my interpretation is correct this should be highlighted as a result and mentioned and discussed afterwards, for the sake of the hypothesis.**

Response: As suggested by the Reviewer, we added a direct comparison of the differences between MW and MB regarding slow-wave properties. To do so, we split the former Figure 4 in two new Figures (new Figure 4 and 5). The new Figure 4 focuses on the impact of mental states on slow-wave properties (see also Figure R2 below) and includes the MB vs MW contrast. Figure R2 shows a cluster of parietal electrodes with larger slopes for MB compared to MW and a small cluster of frontal electrodes with smaller amplitude for MB compared to MW. Despite a tendency for an increase in slow wave density, amplitude and downward slopes over posterior electrodes for MB, we did not observe significant clusters for these parameters over these electrodes. We revised the Results section to include these new analyses (p. 14, l. 279-282):

“Finally, a direct contrast between MB and MW (Figure 4c) indicated that a reduction of slow wave amplitude over frontal electrodes but an increase in their upward slope over parietal electrodes were predictive of MB. No significant clusters were obtained for slow wave density and downward slope.”

Figure R2 (Figure 4). Local properties of slow waves are predictive of mental states

Locally, based on each individual electrode, we performed mixed-effect analyses, following with permutation analysis to quantify the impact of slow wave properties on mental states. 1st column: Density; 2nd: Amplitude; 3rd: Downward Slope (D-slope); 4th: Upward Slope (U-slope). These slow-wave properties were extracted for each electrode and used to compute the t-values (shown in each topographical plot) from t-tests on the following comparisons: (a) MW > ON (b) MB > ON and (c) MB > MW. Black dots denote significant clusters of electrodes ($p_{cluster} < 0.05$ corrected for 12 comparisons using a Bonferroni approach, see Online Methods).

These new Figures and analyses do show reliable differences between MW and MB, which we highlight in the Discussion (see p. 22 and below).

This pattern of reporting is repeated in page 13 lines 244-255, where the association between false alarms and misses, and in particular the association of local sleep amp to MB and MW

254-255 “local sleep amplitude and slope was larger for MW compared to ON reports in frontal electrodes only whereas local sleep amplitude was larger for MB compared to ON reports in posterior channels only ($p_{cluster} < 0.05$, Bonferroni corrected cluster threshold; Figure 4b).”

So, again, **no reporting of lack of diff or not enough evidence for diff between MB and MW**. To be able to claim strongly the differences between the neural markers of local sleep preceding the lapses **a direct statistical comparison between MB and MW would show directly the claims that are described later** in the discussion and in the abstract. “Furthermore, the location of local sleep distinguished sluggish versus impulsive behaviours, mind wandering versus mind blanking.” I think it is important to differentiate a clearly difference to the focus of the task between the types of lapses (MB vs ON and MW vs ON) to a true differentiation between lapses (MB vs MW) in local sleep markers and its associations. Please comment and correct me if I am wrong in my interpretation of the results.

Response: We now show the MB vs. MW contrast for all our analyses : misses and false alarms (p. 9), RT (p. 10), pupil size and vigilance ratings (p. 11), and slow waves (p. 14 and new Figure 4). In particular, we did find statistical differences between MB and MW regarding the properties of slow waves (see above). We modified our Discussion accordingly (p. 22):

“At the behavioural level, we observed that sluggish responses (slow responses and misses) were associated with an increase in slow waves over posterior electrodes (Figure 5a,c). Conversely, impulsive responses (hasty responses and false alarms) were associated with an increase in slow waves over frontal electrodes (Figure 5a-b). At the phenomenological level, compared to task-focused (ON) reports, mind wandering (MW) was preceded by an increase in slow-wave density, amplitude, upward and downward slopes over frontal electrodes (Figure 4a). Compared to ON reports, mind blanking (MB) was also preceded by an increase in slow-wave density over frontal electrodes, but we additionally observed an increase in slow-wave upward and downward slopes over posterior electrodes (Figure 4b). A direct comparison between MW and MB reports shows that MB reports were associated with slow waves with steeper upward slopes over posterior electrodes and smaller amplitude over frontal electrodes (Figure 4c).”

I think it is great to do the modelling in this case as it helps differentiate between go and no go responses in a principled manner in terms of processes happening and possible underlying mechanisms. Yet again, **the results are not directly linked to the type of lapse (MB or MW)** but to the electrode space (in evidence accumulation I mean) and behavioural parameters. So this is a cool result in its on right and happily complementary to a paper we have under review somewhere else (<https://www.biorxiv.org/content/10.1101/2020.07.23.218727v1> no need to cite, just flagging cause there are no other papers on DDHM and arousal), but does not speak directly to the lapse types unless you link it.

Response: We thank the Reviewer for this suggestion and we now report the analysis of the effect of mental states on HDDM parameters in a new Supplementary Figure 7 (see Figure R3 below). Differences between MW and MB, with MW showing a lower threshold and higher decision bias, which we now report (p. 17, l. 336-341):

“Considering the trials that were within 20 seconds from the onset of the probes, we first estimated these different parameters of the DDM for each mental state (Figure S7). This analysis shows differences between MW and MB reports in terms of decision bias and threshold (Figure S7). The lower threshold and higher decision bias observed for MW (compared to MB) are concordant with the idea that MW is associated with behavioural impulsivity.”
These differences could explain the shorter RT observed in MW compared to MB.

Figure R3 (Supplementary Figure 7). Impact of mental states on HDDM parameters.

Hierarchical Drift Diffusion Modelling (HDDM, see Methods) was applied to the Reaction Times obtained in the Face (top) and Digit (bottom) SART. The parameters were fitted for each task and mental state. Each panel shows the distribution of the estimated variable for individual participants (drift for NoGo and Go trials, drift bias, threshold, non-decision time (NDT) and decision bias; see Methods). For each plot, coloured areas show the smoothed distribution of individual data points (see Methods). Diamonds and circles show individual estimates for the Face and Digit SART respectively. Box plots show the 1st and 3rd quartiles (edges) as well as the median (middle bar). Blue boxes around individual plots and stars next to the titles indicate variables with significant state-effects (model comparison, see Methods; *: $p < 0.05$, Bonferroni correction for 12 comparisons). Significant differences between MW and MB (for threshold and decision bias) are highlighted with red brackets.

I will wait for the response to these comments to decide to further engage in details of interpretation and sleep and consciousness frameworks. But I highlight a few papers that maybe will be useful for the discussion about content of the mind in lapses of attention related to sleep onset or fatigue.

Other interesting paper related to content in Sleep onset is "EEG Power Associated with Early Sleep Onset Images Differing in Sensory" where aspect of local sleep could be derived from its fine reading

Content http://dreamscience.ca/en/documents/publications/2001_Nielsen_Germain_Reprint_SRO_4_83-90_EEG_power_at_Sleep_Onset.pdf

Such as changes in local theta and delta associated with specific content.

A study of interest that may be informative is the single case repeated mind wandering in the transition to sleep that we published with Prof Valdas Noreika (Noreika et al 2015). In that paper we show how more the participant trained himself in come out of the transition when the hypnagogic content was interrupted by a dream like bizarreness event or object. “ In particular, an increase in the frequency of Hori stages 4 and 5, and a decrease in the frequency of Hori stage 2 were observed in the last 20 s before a button press (report of hypnagogic event), suggesting that the occurrence of unpredictable hypnagogic experiences may depend on a rapid increase of drowsiness.” The key issue here is that hori 4 is defined by flattening of the EEG and hori 5 by theta ripples, also called sleep-like slow waves of N1 (Hori refs

Hori, T., Hayashi, M., and Morikawa, T. (1994). “Topographical EEG changes and hypnagogic experience,” in *Sleep Onset: Normal and Abnormal Processes*, eds R. D. Ogilvie and J. R. Harsh (Washington, DC: American Psychological Association), 237–253. doi: 10.1037/10166-014

Hayashi, M., Kato, K., and Hori, T. (1999). Hypnagogic imagery and EEG activity. *Percept. Mot. Skills* 88, 676–678. doi: 10.2466/pms.1999.88.2.676

The other result from the 2015 paper that it is useful for this manuscript is the change in theta (slow wave-like) power before the phenomenological report, that differed between types of hypnagogic experiences (linguistic vs visual/contextual).

“Linguistic intrusions were associated with the lower theta power than perceptual imagery reports in the 60–40 s [$t(54) = 3.43$, $p < 0.005$, $d = 1.04$] and 40–20 s [$t(54) = 5.7$, $p < 0.0005$, $d = 1.7$] time windows (see Figure 3A). Furthermore, theta power increased for the linguistic intrusions in the last 20 s compared to 60–40 s [$t(43) = 2.47$, $p < 0.05$, $d = 0.38$] and 40–20 s [$t(43) = 3.07$, $p < 0.005$, $d = 0.45$] time windows. Contrary to this, theta power decreased for the perceptual imagery reports in the last 20 s compared to the 40–20 s time window [$t(11) = 3.33$, $p < 0.05$, $d = 0.96$].”

Response: We are very grateful for these useful references that we have integrated to our Discussion. The hypnagogic period is certainly interesting to put our findings into perspective. Indeed, the physiological, behavioural and phenomenological changes observed during hypnagogia (and late Hori stages) could represent intensified versions of the local sleep events we report here. The fact that, during hypnagogia, individuals can experience vivid subjective experiences (Hayashi et al. *Perceptual and Motor Skills* 1999, Ogilvie *Sleep Medicine Reviews* 2001) inspired our studies as it shows that the transition from wakefulness to early NREM sleep is not necessarily associated with a linear decrease or loss of awareness. In the future, we would like to extend our framework to hypnagogia, to show that hypnagogic reports can also be predicted by the timing and location of sleep-like rhythms. We now mention these papers and considerations in a new paragraph of the Discussion (p. 24-25, l. 494-502):

“Other than mind wandering and blanking, we foresee that the spatio-temporal properties of slow waves might predict other types of spontaneous experiences. For example, previous research focusing on the hypnagogic period at sleep onset has shown that slow-wave like activities are predictive of the occurrence of spontaneous imagery⁷³, the intensity of thoughts⁵⁷ and can discriminate between different contents of spontaneous experiences⁷⁴. Likewise, during sleep, it has been reported that local modulation of slow wave power is predictive of the occurrence and contents of dreams⁷⁵. Local sleep, therefore, might be related to the type and content of spontaneous experiences not only during wakeful states, as we showed here, but also during sleep-wake transitions and sleep. ”

Tristan bekinschtein

Reviewer #2 (Remarks to the Author):

Review of Andrillon and colleagues.

This paper presents an interesting and important study that examines the relationship between EEG patterns during a demanding task context (the SART) and the occurrence of different mental states (On Task, Mind-wandering and Mind-blanking). Using an algorithm that detected high amplitude delta waves in the EEG spectrum, the authors found that these predicted both behaviour and subjective reports of different states. This suggests that high amplitude gamma activity often seen during sleep is an important marker for states of lower vigilance.

Response: We thank the Reviewer for this positive appraisal of our work. We assume that the reference to the “high amplitude gamma activity” was a typo and that the Reviewer meant “high amplitude delta activity”. (Following the Reviewer’s usage, we have replaced “mind state” into “mental state” throughout in the revised manuscript).

I think this is a well-motivated study, that is generally well conducted and that has very important implications for understanding patterns of ongoing thought. There are a few places in which the authors need a more updated account of contemporary accounts of ongoing thought.

Response: Thanks for the suggestion. We have now updated our manuscript to better reflect the current literature (see below for more details).

However, the biggest issue with the paper as it stands is that its argument is somewhat circular and this framing is unfortunately likely to confuse future research moving forward. The notion that **the pattern identified by the algorithm is detecting local sleep, and consequently that all experiences linked to this pattern are therefore an example of ‘local sleep’ is a reverse inference**, and one that the authors should not make the centre of their argument. To play devil’s advocate the data in this paper could also be used to question the specificity of the algorithm for detecting sleep per se given that it also predicts other experiential states. In process terms, this pattern may simply reflect the role that decoupling when we sleep or when we shift focus away from the task (Smallwood, 2013, Psych Bulletin). I am not strongly against the argument that the authors propose because the SART is a really boring task, but the reality is that **there are lots of different ways that these data could be explained** rather than by assuming that because a pattern we see when we sleep predicts a particular state then that state must involve falling asleep.

Instead, why not report what was found: **large amplitude delta waves** are important predictors of an individual’s experiential state when it is not focused on a demanding external task. This is an entirely unambiguous description of the results and it completely removes the circularity that is present in the current manuscript. It may well be that local sleep is linked to some features of ongoing cognition, and I think this is a great question that the authors could examine in subsequent studies, but to my mind nothing in this current paper supports the use of language of detecting local sleep etc... that occurs throughout the paper. Instead it would be easy to use an argument that **a marker that is present in sleep was applied to vigilance data and found to make meaningful predictions of attentional states, thus removing the notion of local sleep from the title and the bulk of the paper and minimise reverse inference**. They can then discuss **why this marker predicts attentional states in the discussion**. I would like to make it clear that in my opinion that this aspect of the language does not take away from the fact that (i) the study is well conducted, (ii) the results are important. I actually think that changing the authors’ language is likely to make this study more important in the future because it opens up the possibility that they have discovered that high amplitude delta waves (sleep related or not) may become a powerful way to understand certain mental states.

Response: Following these remarks, we have made major revisions to the manuscript to clarify our reasoning and more explicitly distinguish between slow waves and their link to local sleep events. We realized that the interchangeable use of the terms “slow waves” and “local sleep events” was causing confusion. In the revision, we clearly distinguish them. Following the Reviewer’s comment on circular augmentation and reverse inference, we have thoroughly revised the manuscript in four ways as we describe in the following.

First, we now also clarify and justify our methodological approach and make a clear distinction between the slow waves we measured and our interpretation of local sleep. Specifically, when we observe “slow waves” locally and in wakefulness, we interpret them as a marker of “local sleep” (p. 11, l. 221-226):

“To further examine the potential mechanistic link between sleep pressure and attentional lapses, we set out to detect a marker of sleep pressure in the EEG signal, in the form of local, sleep-like slow waves. We recently reviewed the rationale behind this approach¹¹. In particular, relying on a local marker of sleep allowed us to test whether distinct families of attentional lapses can be coherently explained by the occurrence and spatio-temporal characteristics of slow waves.”

Throughout the manuscript (including the title itself), we now clearly distinguish observation and interpretation. For example, p. 20-21, l. 405-410:

“We propose that these two types of attentional lapses can be explained by the occurrence of sleep-like low-frequency high-amplitude waves, where regional differences in the occurrence of slow waves predicts the type of attentional lapse as well as its behavioural profile. These slow waves have been previously linked to the concept of local sleep^{11,22,25,29,30} and, accordingly, we interpret here the presence of slow waves as markers of local sleep.”

Importantly, the significance of the results we report go beyond their interpretation as local sleep, as pointed out also by Reviewer 3.

Second, as to the issue of reverse inference, we decided to provide a coherent set of evidence to strengthen our argument. We do agree with Reviewer that cognitive neuroscientists should be careful not to draw quick conclusions based on reverse inference (Poldrack *TICS* 2006). Nonetheless, when carefully employed, especially with convergent evidence, reverse inference can be regarded as an effective and empirical strategy for cognitive neuroscientists (Hutlzer *Neuroimage* 2014).

In this spirit, the link we propose between MW and MB and the phenomenon of local sleep does not rest on a single reverse inference from the literature but on an array of converging evidence from EEG, pupilometry, behaviour and subjective reports. Further, in the revision, we added three new elements to this array of evidence. (1) We improved the comparison of slow waves between wake and sleep (new Figure 3). (2) We report a positive correlation between slow waves and pupil size (see p. 13, l. 257-265). (3) Importantly, we also showed that slow waves increase with time spent on task (see p. 13, l. 263-265 and new Figure S1), matching a known property of local sleep (Vyazovskiy et al. *Nature* 2011; Hung et al. *Sleep* 2012; Bernardi et al. *J Neuroscience* 2015). We now dedicate a new paragraph in the Discussion on the relationship between slow waves and local sleep (p. 25-26, l. 524-535):

“We speculate that the slow waves we report here are generated by similar neural mechanisms as slow waves in sleep. Our speculation rests on the profile of slow waves characterized in (i) time (Figure 3a) and (ii) space (Figure 3b,c), and the relationships between slow wave properties and (iii) time spent on task (Figure S1), (iv) subjective vigilance and (v) pupil size. While these five lines of evidence are correlational, we note that together, they imply a high degree of similarity between slow waves in waking and sleep. This, in turn, allows us to interpret wake slow waves as local sleep. Furthermore, we showed that slow waves differ from typical responses evoked by stimuli or participants’ responses (Figure S9). Understanding the exact neural mechanisms that generate slow waves would require direct intracranial recording in humans²¹. Such invasive recordings would ensure that these slow waves are accompanied by episodes of neuronal silencing, as for sleep slow waves.”

Third, following the Reviewer’s suggestion, we now discuss the relationship between slow waves and the notion of sensory decoupling (p. 23, l. 460-467):

“Based on our results, we speculate that local sleep plays a key role in the complex relationship between the DMN and spontaneous experiences. Previous studies have indeed shown that a state of low alertness could induce the phasic activation of the DMN⁵³. We speculate that slow waves occurring within the DMN could lead to episodes of mind wandering by disconnecting individuals from their environment⁵⁴. Indeed, slow waves during sleep have been proposed to be responsible for disconnecting sleepers from their environment^{27,55}, as they are accompanied by a phenomenon of neuronal silencing that can disrupt the processing of external inputs.”

Thus, the local sleep interpretation provides a mechanistic explanation for the association between mind wandering and sensory decoupling (i.e. because mind wandering could be associated with slow waves and neuronal silencing). Following the distinctions proposed in Smallwood, *Psych Bulletin* (2013), we also further develop the idea that local sleep could offer a mechanistic account of the processes that “control the occurrence” of mind wandering but does not necessarily speak to the “processes that ensure the continuity” of mind wandering once initiated (p. 23, l. 467-468):

“If slow waves trigger specific occurrences of mind wandering, other mechanisms could be responsible for the stability of these episodes⁵⁶.”

Finally, to buttress our local sleep interpretation, we now also explore alternative explanations as suggested by the Reviewer and we distinguish slow waves from task-related events (stimulus and motor responses; see Figure R8 and Discussion p. 26, l. 530-532).

20 second window for evaluating the links between brain and behaviour. The authors use a 20 second window to examine experiential states, however, while this has been used in the past (i.e. Christoff et al., 2009) this is in fact a rather long window and **they should justify why they used such a long one to predict experience when EEG can resolve things much faster than fMRI can**. In work from my lab we have employed 6 second windows successfully (Turnbull et al., 2019, Nat Comms; Sormaz et al., 2018). We do this because shorter windows have the advantage of being less ambiguously linked to subsequent experiential reports rather than things that systematically precede it (like local sleep for example). This is especially important because we know that changes over this length of time influence the likelihood of reporting off task states (e.g. Turnbull et al., 2018, Neuroimage) and that over a similar time frame changes in neural activity also occurs and that this is predictive of reports of off task thought (Turnbull et al., 2020, Scientific Reports). **To address this it may be possible for the authors to use a shorter window**, or based on the evidence I just described, evaluate whether changes over this period matter for the reports that participants subsequently give.

Response: Thanks for the suggestion. Following the Reviewer’s comment, we reanalysed our behavioural and EEG data with a shorter time window. In general, the results were consistent with (though sometimes weaker than) the original findings. We report this in the revised manuscript.

For behavioural results, we focused on windows of 10s prior to probes, so that these windows would contain at least 1 NoGo trial. Table R2 shows the behavioural analysis obtained for these 10s-long windows (right column), compared to the original 20s-long windows (left column). We now report this new analysis in our revised manuscript (p. 10, l. 201-203).

		20s before Probes	10s before Probes
Misses (%) in Go trials	State	$\chi^2(2)=36.0$ (***)	$\chi^2(2)=31.3$ (***)
	MW vs ON	-0.011 [-0.016, -0.0052]	-0.012 [-0.019, -0.0037]
	MB vs ON	-0.023 [-0.032, -0.015]	-0.033 [-0.045, -0.021]
	MB vs MW	-0.013 [-0.021, -0.0047]	-0.021 [-0.033, -0.0096]
False alarms (%) in NoGo trials	State	$\chi^2(2)=115.9$ (***)	$\chi^2(2)=55.97$ (***)
	MW vs ON	-0.20 [-0.24, -0.17]	-0.20 [-0.25, -0.15]
	MB vs ON	-0.17 [-0.23, -0.12]	-0.14 [-0.22, -0.062]
	MB vs MW	0.028 [-0.028, 0.084]	0.061 [-0.018, 0.14]
RT (s) in Go trials	State	$\chi^2(2)=16.9$ (**)	$\chi^2(2)=8.42$ (ns)
	MW vs ON	-0.0025 [-0.0096, 0.0045]	0.0026 [-0.0066, 0.012]
	MB vs ON	0.019 [0.0088, 0.030]	0.020 [0.0064, 0.034]

	MB vs MW	0.022 [0.011, 0.032]	0.018 [0.0038, 0.031]
--	-----------------	-----------------------------	------------------------------

Table R2 (Supplementary Table 3). Effect of different pre-probe window sizes on behavioural analyses

The analyses of the effect of mental states (model comparisons and post-hoc contrasts) were replicated using a shorter window of 10s (right column) instead of 20s before probe onsets (left column). Likelihood Ratio Tests are reported for the effect of State on the correctness for Go and NoGo trials as well as the Reaction Times (RT) for Go Trials. Stars denote the corrected significance level (Bonferroni correction for 6 comparisons: ***: $p < 0.001$, **: $p < 0.01$, *: $p < 0.05$). The estimates for the post-hoc contrasts and the corresponding 95% confidence intervals are also shown. Bold fonts signal the significant results ($p < 0.05$ for model comparisons, or confidence intervals excluding 0 for the post-hoc contrasts)

For EEG analyses, Figures R4-6 show the slow wave analyses in relation to the mental states, breaking the initial 20s analysis into four 5s windows. With these analyses, we note that (1) the cluster of electrodes predictive of mental states tend to appear close to the probe onset, (2) clusters for the MW vs ON contrast tend to be centro-frontal while clusters for the MB vs ON contrast tend to be posterior. Both observations buttress our initial results. We now report this analysis in the Discussion (p. 22, l. 438-441) and the Result section (p. 14, l. 283-289):

“To provide more details on the temporal relationship between slow waves and mental states, we replicated this analysis by splitting the 20s window before probes into four windows of 5s (Figure S2-4). The contrasts between MW and ON (Figure S2) as well as MB and ON (Figure S3) show that the properties of slow waves best predict mental states within the 5s before a probe. In terms of topography, when compared to ON state, MW is best predicted by the slow wave properties in frontal electrodes (Figure S2) while MB is best predicted by those in the centro-parietal electrodes (Figure S3).”

Figure R4 (Supplementary Figure 2). Spatiotemporal dynamics of the effect of slow-waves properties on mental states (MW vs ON).

Mixed-Effects Models were used to quantify the impact of slow-wave properties (a: Density; b: Amplitude; c: Downward Slope (D-Slope); d: Upward Slope (U-Slope)) on mental states as in Figure 4. Slow waves were detected on 4 different windows: [-20, -15]s, [-15, -10]s, [-10, -5]s and [-5, 0]s before probe onsets. Slow-wave properties were extracted for each electrode and used to predict the MW vs ON contrast. Topographies show the scalp distribution of the t-values associated with each slow-wave parameter and electrode. Black dots denote significant clusters of electrodes ($p_{\text{cluster}} < 0.05$ corrected for 48 comparisons (Figure S2-4) using a Bonferroni approach).

Figure R5 (Supplementary Figure 3). Spatiotemporal dynamics of the effect of slow-waves properties on mental states (MB vs ON). Same format as in Supplementary Figure 2.

Figure R6 (Supplementary Figure 4). Spatiotemporal dynamics of the effect of slow-waves properties on mental states (MB vs MW).

Same format as in Supplementary Figure 2.

Discussion of the neural correlates of off-task thought is too simplistic. Although initial views argued that the DMN supports off task thought, we now know that this is an oversimplification and the authors should try to reflect this more nuanced view in the discussion. First, using multiple experience sampling questions in a data driven manner we know now that being off-task is related to reductions in task positive systems, such as the dorsal attention network (Turnbull et al., 2019). It is true that activity in the DMN can be important in the off-task content, e.g. Konu et al., (2020, Neuroimage) found activity in vmPFC was linked to the expression of off task social content. However, using a similar experience sampling protocol we found using both representational similarity analysis (Sormaz et al., 2018, PNAS) and standard whole brain uni variate analysis (Turnbull et al., 2018, Neuroimage, 2019, Nat Comms) that the DMN is important when people report experiences with task focused detail in a working memory task. This effect is likely to be targeted on the posterior cingulate cortex (Turnbull et al., 2019) and we suspect it may serve this function even in tasks that are default mode network friendly, such as self-reference (Murphy et al., 2019, PLOS One). Complicating things further regions like the dLPFC can also play a role in off task states (Turnbull et al., 2019) perhaps playing a role in the prioritisation of information relevant to the individuals goal state. **It would be useful to try to incorporate the complexity that these studies raise in a revision, perhaps commenting on how their results, and the application of EEG, could help move forward the question of the role of the DMN in cognition.** For example, it struck me when reading this paper whether when there are high amplitude delta waves dominate the default mode network becomes decoupled from the task which could explain why it can play a role in both task relevant and off task states.

Response: Thanks for your suggestion. We have now refined our discussion on the DMN. We agree with the Reviewer that our Results provide some novel insights on the role of the DMN in spontaneous experiences. We extended our treatment of this matter in the Discussion as follows (p. 22-23, l. 450-471):
“Early fMRI studies showed that mind wandering in this sense was associated with the activation of the Default Mode Network (DMN)^{45,46}. Relevant to our local sleep interpretation, the DMN is also suggested to be involved in dream generation during sleep, consistent with the idea that the DMN supports a broad array of experiences that are decoupled from the environment⁴⁷. Furthermore, lesions within the DMN are associated with a decrease in both mind wandering⁴⁸ and dreaming⁴⁹. However, recent findings suggest a complex relationship between the DMN and spontaneous experiences. Unlike what was initially hypothesized, the DMN is now considered to be involved in both task-unrelated and task-related processes^{43,50}, depending for example on environmental demands or the vividness of individuals’ experiences^{51,52}. Based on our results, we speculate that local sleep plays a key role in the complex relationship between the DMN and spontaneous experiences. Previous studies have indeed shown that a state of low alertness could induce the phasic activation of the DMN⁵³. We speculate that slow waves occurring within the DMN could lead to episodes of mind wandering by disconnecting individuals from their environment⁵⁴. Indeed, slow waves during sleep have been proposed to be responsible for disconnecting sleepers from their environment^{27,55}, as they are accompanied by a phenomenon of neuronal silencing that can disrupt the processing of external inputs. If slow waves trigger specific occurrences of mind wandering, other mechanisms could be responsible for the stability of these episodes⁵⁶. Further investigations, including source localization or simultaneous recording of EEG and fMRI^{57,58}, promise a deeper understanding of the mechanisms underlying these attentional lapses.”

Signed

Jonathan Smallwood

Minor issues

It might make sense in the discussion to explain that the use of the SART as a paradigm may have contributed to some of the patterns the authors observe. This is because the SART requires participants maintain attention for prolonged amount of time with minimal external support and we know that the neural correlates of both on tasks and off task states change when tasks are designed so that they have minimal vigilance load (e.g. the 0 – back / 1 back paradigms used by Sormaz et al., etc...). The use of multiple tasks conditions in studies like this is helpful because it provides clarity on the contextual conditions within which we might expect the results to emerge (Smallwood and Andrews-Hanna, 2013). Since this was not done in this study it should be mentioned. This is not a limitation because power is lost when the experiment is divided into multiple conditions but it is nonetheless important to mention it because we don’t know the contextual specificity of the effect yet.

Response: We agree with the Reviewer. We added the following caveat to the Discussion (p. 26-27, l. 536-539):

“It is worth noting that the tasks used here (SARTs) are rather undemanding and could favour sleepiness and local sleep compared to more difficult and engaging experimental paradigms. Generalising our findings to different experimental contexts or more naturalistic settings would need further experimental validation¹¹.”

Page 7. “Taken together, these results suggest that MW and MB can decrease performance through different ways: MW facilitates impulsivity, evidenced by faster RT and more false alarms, whereas MB facilitates sluggishness, evidenced by slower RT and more misses.”

Unless I misunderstood the analysis, could this also indicate that MB is fully disengaged when MW was only partially disengaged? (i.e. it was not faster than on task and MB was worse than MW?).

Response: We apologise if this description of the results lacked clarity. We rewrote the results section as follows (p. 10, l. 192-198):

“Taken together, these results suggest that MW and MB decrease performance in different ways. MB is characterized with more misses and slower RT than MW and ON, which is consistent with the idea that MB induces sluggish mental states. As to MW, the analysis of misses may ostensibly be compatible with an idea of MW as a mild form of MB, however, the fact that RT in MW is faster than in MB is not easy to reconcile with such an idea. Instead, this overall pattern is consistent with an idea that MW induces more impulsive mental states.”

Page 8 and the Discussion on pupil dilation. The author may be interested to note that Konishi and colleagues (2017, Cognition) also found that on tasks states had heightened pupils relative to off task states but only those with a temporal focus on the past, or where highly intrusive in nature.

Response: **We thank the Reviewer for this reference, which we added to the manuscript.**

Reviewer #3 (Remarks to the Author):

In this study Andrillon and colleagues investigated the relationship between local sleep, mental states and performance on a sustained attention to response task (SART) in 26 healthy awake individuals. They report that the occurrence of large EEG slow waves ('local sleep') is not only associated, in a region specific manner, with errors in the task (hits vs. misses), but also with distinct mental states (mind blanking vs. wandering).

The existence of local sleep in wakefulness and its regionally specific effects on behavior have been reported previously in both humans and rodents. **What is new and noteworthy here is the finding that large EEG slow waves during wakefulness are associated with reports of mind blanking/wandering in a regionally specific manner.** If confirmed, this finding would be **of interest for (neuro)science in general, beyond the field of sleep.**

Response: **We thank the Reviewer for this positive assessment and appreciation of our work.**

My main concern with this paper is that it **lacks the crucial proof that the large EEG slow waves they investigate are truly sleep related.** Such a proof would require the demonstration that mindblinking/wandering and the associated EEG slow waves increase as **a function of time spent awake** (that is, as a function of homeostatic sleep pressure), or **intracranial recordings documenting neuronal off-states** typical of sleep slow waves, as has been done in previous studies. Here the authors use correlations with subjective vigilance scores and pupil size as a proxy, which in my view is insufficient to sustain such a strong claim. Neither EEG low-frequency oscillations nor pupil size are unambiguous markers of sleepiness/local sleep.

Response: **We agree with the Reviewer that caution is required in interpreting our results. Based on these comments and those of the other reviewers, we have made major revisions to the manuscript. Specifically, we have worked hard to emphasize that beyond the local sleep interpretation, there is a broader impact in our finding. We have also clarified our reasoning and provided additional evidence in support of the local sleep interpretation.**

First, the overarching goal of this paper was not to decisively prove that wake slow waves are truly sleep related. This would require, as pointed out by the Reviewer, intracranial recordings to link slow waves with episodes of neuronal silencing (as done in rodents in Vyazovskiy et al. Nature 2011) and/or long-term EEG recordings of both wake and sleep to show that slow waves are homeostatically regulated (i.e. increase with time spent awake and decrease with time spent asleep). In this paper, we could not take these approaches. Instead, we relied on an array of converging evidence that supports the link between the slow waves we detected and the concept of local sleep, as defined and commonly measured in the literature. This was precisely because our main objective was to link behavioural and neural measures with subjective experience of MW and MB. Consequently, our study design focused on healthy participants equipped with high-density EEG to optimise the quantity and quality of reports about subjective experience, allowing us to establish a link between slow waves, participants' behaviour and subjective experience than other designs. While we agree with the reviewer that extending our studies to intracranial recordings in epilepsy patients or sleep deprivation protocol with healthy patients would strengthen the link between behaviour and neural measures, such studies would have made it more difficult to establish the link between these measures with subjective experience of MW and MB. Nonetheless, these are avenues of research we will explore in the future.

Second, we now give stronger evidence in favour of the local sleep interpretation. The evidence that we presented in the first submission included: (1) the negative correlation between slow waves and pupil size, and (2) the positive correlation between slow waves and sleepiness ratings. In this revision, we now include the following novel analyses: (3) the demonstration that slow-wave density increases with the time spent on the task (Figure R7 below), (4) the close resemblance of the topography of slow-wave density and amplitude in wake and sleep (Fig. R11) and (5) the clear differences in the waveforms between slow waves, stimulus-evoked, and response-evoked potentials (Fig. R8). We hope these convergent lines of evidence are convincing enough to link slow waves and local sleep to behaviour and subjective experience. We now dedicate a specific paragraph, in the Discussion, to clarify our argumentation and its supporting evidence (p. 25-26, l. 524-535):

"We speculate that the slow waves we report here are generated by similar neural mechanisms as slow waves in sleep. Our speculation rests on the profile of slow waves characterized in (i) time (Figure 3a) and (ii) space (Figure

3b,c), and the relationships between slow wave properties and (iii) time spent on task (Figure S1), (iv) subjective vigilance and (v) pupil size. While these five lines of evidence are correlational, we note that together, they imply a high degree of similarity between slow waves in waking and sleep. This, in turn, allows us to interpret wake slow waves as local sleep. Furthermore, we showed that slow waves differ from typical responses evoked by stimuli or participants' responses (Figure S9). Understanding the exact neural mechanisms that generate slow waves would require direct intracranial recording in humans²¹. Such invasive recordings would ensure that these slow waves are accompanied by episodes of neuronal silencing, as for sleep slow waves."

Finally, the relevance of our findings on the regional distribution of slow waves in relation to MW and MB does not depend entirely on the local sleep interpretation. Specifically, we show here that high-amplitude slow waves, which correlate with objective and subjective indexes of sleepiness, precede the occurrence of mind wandering and mind blanking. In addition, their spatio-temporal properties differentiate between mind wandering and mind blanking, impulsive and sluggish behaviour. These novel and specific results could shape future studies, within or beyond the sleep field, which may benefit from the prediction of mind wandering and blanking. Applications for brain machine interfaces in the context of education or in the workplace could leverage our results as well. We briefly make this point in Discussion (p. 27, l. 549-551): "Identifying a proximate mechanism of attentional lapses could inspire novel applications leveraging brain-machine interfaces in educational or professional settings."

Figure R7 (Supplementary Figure 1). Slow waves increase with time spent on task

(a) Topographies of the temporal density of slow waves (slow waves per minute, averaged across all 26 subjects). An increase in the number of slow waves, maximal over central electrodes, can be seen from the first to the last experimental block. (b-d) Slow wave density for electrode Cz (b), vigilance ratings (c) and pupil size (d) averaged within each block. Connected larger dots show the average across participants (error-bars: SEM). Individual semi-transparent circles show individual subjects.

Other comments:

2) The EEG recording used to study the association between slow waves and mind states appears to contain both stimulus-induced and spontaneous activity. This is an issue, as large stimulus-evoked potentials can sometimes be observed in the wake EEG and cannot be considered 'local sleep'. Although the large waveform shown in Figure 1a represents an average of 26 subjects, it appears extremely sharp and is preceded by a fast and clear positive-negative-positive deflection – could this reflect such stimulus-evoked activity?

Response: We thank the Reviewer for this comment. It is indeed important to check that the slow waves we detected and reported are not trivially linked to task evoked activity. To show this, we compared the average ERP waveform (see new Figure R8 below) to event-related potentials locked to stimuli onsets (for the digit and face stimuli separately) and to participants' manual responses. These ERPs are displayed for electrode Cz but we also show the topographies of the ERPs at the time of the peak for Cz. As shown in Figure R8a-c, the detected slow waves critically differ from task-related ERPs in the magnitude of the peak and the presence of a large negative trough across all electrodes. This is particularly important as it is this negative trough that has been associated with neuronal silencing. We also plot, for one subject, the voltage values of single-trial task-related ERPs and slow waves, which show that slow waves are typically much larger than single-trial responses to task-events (Figure R8d-f). It therefore appears that the detected slow waves do not reflect stimulus-evoked activity.

Figure R8 (Supplementary Figure 9). Slow waves compared to stimulus and response-locked activity
 Event-related potentials averaged over electrode Cz and across participants for slow waves (a), stimulus-locked activity (b) and response-locked activity (c). Slow waves' EEG time courses are aligned with the first negative crossing before the negative peak. Stimulus-locked responses are aligned on the onset of the digit (full line) or face (dotted line) stimuli. Response-locked responses are aligned to the onset of participants' responses. Shaded areas show the SEM across participants (N=26). Insets show the topography of the average voltage at the times shown by the vertical dotted lines (a: 0.05s; b: 0.2s; c: -0.05s), which approximates the absolute maximum of each ERP. (d-f) Voltage of the event-related potentials per trial for one participant. Each row corresponds to (d) an individual slow wave, (e) a stimulus presentation (face or digit), (f) a motor response. We limited the number of rows in (e) and (f) to match the number of detected slow waves (N=120) in (d).

3) Methodological choices are not always justified and therefore appear arbitrary at times:

- In Figure 3c, the authors show the distribution of slow wave parameters across different mind states. To quantify the effects, they run a "mixed-effect model analysis focusing on local sleep events before each probe and **across all electrodes**". Why is this "main" effect of mind state (model comparison with likelihood ratio) evaluated on the **average across electrodes** (considering that LOCAL sleep is the main focus) and the contrasts performed electrode by electrode (shown in Figure 4)?

Response: We did not average across electrodes. We rather meant that the analysis was conducted by looking at slow waves detected on all electrodes, with electrodes coded as a fixed effect within the mixed effect model. The mixed-effect models that we were built and tested were the following:

$$X \sim 1 + \text{Task} + \text{Electrode} + (1 | \text{Subject})$$

$$X \sim 1 + \text{Task} + \text{Electrode} + \text{MS} + (1 | \text{Subject})$$

Where X represents slow-wave density or properties.

Having said that, we agree with Reviewer 3 that these analyses could be misleading and distract the readers from the by-electrode analyses looking into the impact of local slow waves on attentional lapses, which address more directly our hypotheses. We therefore removed this paragraph from the results section and now focus more simply on the “electrode by electrode” analysis. The Results section and Supplementary Table 1 have been updated accordingly.

- Why were different time windows prior to probes (20s vs 30s) used for different analyses (lines 480-81)?

Response: We apologise if our explanation of the motivation behind the use of 20s vs 30s analyses was unclear. For all analyses related to mental states, we used 20s prior to probes in order to maximize the temporal proximity of the trials to the subjective reports. For the analyses that did not focus on mental states, we did not need to use data that are close to subjective reports (e.g., when examining the impact of slow waves on behaviour independently of mental states). Consequently, to maximize the power of our analyses, we had extended the window to 30s, which is the minimal inter-probe interval.

However, following the Reviewer’s comment and to simplify our methods, we replicated all our analyses using the same window of 20s. The results were not affected and we updated the manuscript (Methods: p. 30, l. 631-636; p. 33, l. 696-698; Results: p. 16, l. 304) and Figures accordingly (see Figures R9 and R10 below).

Figure R9 (Figure 5). Local occurrences of slow waves are associated with modulations of performance

Mixed-Effects Models were used to quantify the correlation between slow-wave occurrence and reaction times (a), false alarms (b) and misses (c) at the single-trial level. Topographies show the scalp distribution of the associated t-values. Black dots denote significant clusters of electrodes ($p_{\text{cluster}} < 0.05$ corrected for 3 comparisons using a Bonferroni approach, see Online Methods).

Figure R10 (Figure 6). Global and local effects of the occurrence of slow waves on sub-components of decision-making
 Reaction Times in the Go/NoGo tasks were modelled according to a Hierarchical Drift Diffusion Model (see Online Methods). a-f: Topographical maps of the effect of slow waves (i.e. whether or not a slow wave was detected for each trial and for a specific electrode) on the parameters of decision-making: drift go [v_{GO}] (a), drift nogo [v_{NOGO}] (b), drift bias [v_{BIAS}] (c), threshold [a] (d), Non-decision time or NDT [t] (e), decision bias [z] (f). The effect of slow wave occurrence was estimated with LMEs (see Online Methods) and topographies show the scalp distribution of the associated t-values. Black dots denote significant clusters of electrodes ($p_{cluster} < 0.05$, Bonferroni-corrected, see Online Methods).

- Why was a threshold of $p < 0.01$ used for the cluster permutation approach, while for the Monte-Carlo simulations, a threshold of $p < 0.05$ was used?

Response: Our usage of two different thresholds in this statistical method is standard and based on the influential article from Maris and Oostenveld (Journal of Neuroscience Methods, 2007). In short, the first threshold, which we called the “cluster alpha”, defines the cluster candidates that we test in the subsequent permutation test. The Maris and Oostenveld paper is clear on the fact that this threshold is chosen on a case-by-case basis as the goal here is to “Select all samples whose t-value is larger than some threshold” (quotes are from the Maris and Oostenveld paper). Furthermore, the authors add that “This threshold may or may not be based on the sampling distribution of the t-value under the null hypothesis, but this does not affect the validity of the nonparametric test”. Accordingly, the choice of threshold impacts the number and size of the clusters, whose significance is assessed through permutations. The choice of this threshold thus influences the rate of type-2 errors (false negatives), making it difficult to interpret the absence of significant clusters (which we avoided doing in our manuscript). Crucially however, the choice of the cluster alpha does not affect the type-1 error rate (so does not increase the likelihood of falsely reporting a significant cluster). In our case, we initially set this cluster alpha initially to 0.01 but, as we added several new analyses focusing on shorter windows and less data (e.g. Figures S2-4), we changed it to 0.025 for this revision to make it less restrictive.

Once the clusters are identified with the cluster alpha, a permutation approach is used to determine whether the detected clusters are significant compared to random permutations. In this case, a threshold is used (typically 0.05) that will control for the type-1 error rate (the lower this threshold the less likely false positives are). In our case we set this threshold to 0.05. However, since we sometimes replicated this cluster permutation approach for non-independent tests (e.g. examining the different properties of slow waves), we further corrected the cluster significance threshold using a Bonferroni approach. This was done so that we could maintain the same rate of type-1 errors across all permutation analyses.

This is now explained in more detail in the Online Methods (p. 34, l. 722-729):

“In practice, for each electrode, we extracted the t-values and p-values for the effects of interest. Clusters of neighbouring electrodes were defined as electrodes with p-values below 0.025 (cluster alpha). Once the clusters were defined, a comparison was performed with permuted datasets and we used a Monte-Carlo p-value threshold of 0.05 to identify the significant clusters ($p_{cluster}$, see Supplementary Methods for details). This threshold was corrected for multiple comparisons when several non-independent cluster permutations were performed (e.g. Figure 4-6, S2-4) in order to keep the type-1 error rate constant across analyses.”

We also updated the Supplementary Methods (p. 5, l. 100-108) to give further explanations about how clusters are identified and tested:

“Candidate clusters were defined as neighbouring electrodes with a p-value below a threshold (called “cluster alpha”) of 0.025. For each candidate cluster, we computed the sum of the t-values for all the electrodes belonging to the cluster (which we will refer to as the “cluster statistics”). We then created permuted datasets by permuting the labels of the predictor within each subject, each task and each electrode (N=1,000 permutations). For each of these permuted datasets, we also identified the candidate cluster with maximal absolute cluster statistics. The cluster statistics from permutations formed a null distribution, against which we compared the cluster statistics from the real dataset. Clusters (real and permuted) with positive and negative cluster statistics were compared separately.”

- Why were mind blanking and “don’t remember” trials grouped within the same category, considering that “don’t remember” trials are few and may have distinct neurophysiological substrates (see for instance Fazekas Sleep Med Reviews 2018 for an analogy with sleep experiences)?

Response: Grouping mind-blanking (stricto sensu) and “don’t remember” reports matches the methodology of one of the few papers that compared mind wandering and mind blanking (van den Driessche et al. Psych Science 2017: “*mind blanking includes reports of blank thoughts and failures to report existing contents (i.e., failures of metacognition)*”). We initially thought to differentiate these two responses, inspired like the Reviewer by the literature on sleep and dreaming (see also Siclari et al. Nature Neuroscience 2017). But the low number of “don’t remember” reports (1.1%, that is <1 probe per participant) made it impossible to do so. We therefore decided to merge these two responses. We now explain this more clearly (p. 28-29, l. 589-595):

“Participants had to select one of the four following options: (1) “task-focused” (i.e. focusing on the task, ON), (2) “off-task” (i.e. focusing on something other than the task, which we define here as mind wandering MW), (3) “mind blanking” (i.e. focusing on nothing), (4) “don’t remember”. As the 4th option accounted for only 1.1% of all probes (i.e. less than 1 probe per participant on average) and since previous studies do not always distinguish between these options (e.g. 86), we collapsed the 3rd and 4th options as mind-blanking (MB) in all analyses.”
We also checked that our results were not affected by the exclusion of “don’t remember” reports.

- The authors only selected waves with the highest **absolute peak-to-peak amplitude (top 10%)**. **Why did they chose this threshold? How variable was the absolute amplitude of these waves between subjects? Is the incidence of such local sleep events (3-8/min depending on electrodes) comparable to other studies?**

Response: We chose the top 10% to select the upper tail of the distribution based on a visual inspection of the data, which we describe in the revised manuscript (p. 31, l. 659-662):

“This 10% threshold was selected based on the visual examination of the distribution of the amplitudes at the subject level (see Figure S10 for the corresponding topography). The mean \pm SEM of the threshold voltage across subjects was $30.7 \pm 1.5 \mu V$.”

We also added a Supplementary Figure showing the scalp topography of the average and standard deviation (across participants) of the thresholds in μV (Figure R11).

Figure R11 (Supplementary Figure 10). Scalp topography of the average and standard-deviation of the slow-wave detection threshold

Only slow waves whose amplitude is within the top-10% of the distribution for each electrode and participant were analysed. The left topography shows the value, in voltage, of this threshold for each electrode, averaged across participants. The right topography shows the standard deviation (SD) of the threshold values across participants.

Regarding the comparison in the number of slow waves per minute with previous findings, to the best of our knowledge, we don't think that this metric was reported elsewhere with the method that we used. It would certainly be interesting to be able to compare the number of slow waves observed in our non-sleep-deprived subjects with participants undergoing acute sleep deprivation.

- Why were two different tasks performed (face and digit)?

Response: We were originally interested in the perceptual component of the task and how mental states relate to perceptual processes. We decided therefore to vary the nature of stimuli, using both digits and faces. We also thought that two variants of the same task would break the monotony of the task. However, to keep this manuscript to a reasonable length, we decided to centre this study on participants' behaviour and subjective reports and we will analyse the perceptual responses later.

- Why are only false alarms shown in Figure 2 and not misses (lines 137-140)?

Response: We now report Misses in Figure 2a and move the former Figure 2a to Figure 1.

Figure R12 (Figure 2). Low arousal is associated with attentional lapses characterized by different behavioural outcomes

Proportion of misses (a) and false alarms (b) in the 20s preceding ON, MW and MB reports. The markers' size is proportional to the number of reports for each participant (same for c-e). Grey diamonds and circles show the average across participants, weighted by the number of reports (same for d and e). c: Distribution of reaction times (RT) for Go Trials (left: Face; right: Digit) in the 20s preceding ON, MW and MB reports. d: Vigilance scores (subjective ratings provided during probes) associated with ON, MW and MB reports. e: Discretized pupil size (see Online Methods) in the 20s preceding ON, MW and MB reports. In a-e, stars show the level of significance of the effect of mental states (Likelihood Ratio Test, see Online Methods; ***: $p < 0.005$).

4) It is sometimes difficult to follow the authors' conclusions based on the data that is presented:

- Lines 177-179: "Taken together, these results suggest that MW and MB can decrease performance through different ways: MW facilitates impulsivity, evidenced by faster RT and more false alarms, whereas MB facilitates sluggishness, evidenced by slower RT and more misses." This conclusion is not entirely correct because no difference in false alarms was found between MB and MW.

Response: The Reviewer is correct and we corrected the sentence as follows (p. 10, l. 192-198): "Taken together, these results suggest that MW and MB decrease performance in different ways. MB is characterized with more misses and slower RT than MW and ON, which is consistent with the idea that MB induces sluggish mental states. As to MW, the analysis of misses may ostensibly compatible with an idea of MW as a mild

form of MB, however, the fact that RT in MW is faster than in MB is not easy to reconcile with such an idea. Instead, this overall pattern is consistent with an idea that MW induces more impulsive mental states.” We apologize for this mistake.

- Slow wave slope and amplitude are taken as equivalent markers of local sleep (Figure 4b, text lines 252 and following), although their topography does not overlap (ex. MW/On contrast) and without a clear justification. Previous studies have shown that the parameters likely reflect distinct neurophysiological processes. Also, it is not clear why the authors report on downward but not upward slope and whether they assessed slow wave density in this context. Results for all parameters should be reported.

Response: Thanks for noticing this. In the revised manuscript, we now distinguish more clearly between these dimensions of slow waves and report them more explicitly, as you can see in Figure R13.

Figure R13 (Figure 3). Properties of Slow Waves

a: Average waveform of the slow waves detected over electrode Cz during the behavioural tasks (red, left; N=26 participants). The average waveform of slow waves detected during sleep (blue, right), extracted from another dataset (see Supplementary Methods), is shown for comparison. Slow waves were aligned by their start, defined as the first zero-crossing before the negative peak (see Online Methods). b: Scalp topographies of the density of slow waves (arbitrary units) detected in wakefulness (top) and sleep (bottom). c: Scalp topographies of wake slow-waves properties (1st column: temporal Density; 2nd: peak-to-peak Amplitude; 3rd: Downward Slope (D-Slope); 4th: Upward Slope (U-Slope); see Online Methods) averaged across participants (N=26). d: Scalp topographies for slow-waves Density (1st column), Amplitude (2nd),

Downward Slope (D-Slope, 3rd) and Upward Slope (U-Slope, 4th) for the different mental state (ON, MW and MB).

- Line 200: "Both the temporal profile and topographical distributions of local sleep detected during the tasks (Figure 3a-b) resemble the slow waves observed in NREM sleep". Please provide the topoplots for sleep, not only the waveform, so that one can directly compare.

Response: We added this information to Figure 3b.

5) Methodological clarifications:

- Please clarify whether the subjects were well rested when doing the task, or whether the experiment was performed in conditions that maximize the occurrence of local sleep (i.e. sleep restriction, specific time of the day, etc.).

Response: Participants were not instructed to follow a specific schedule before the task. The timing of experiments depended on participants' availability. As we did not collect information about participants' sleep/wake cycles prior to the experiment or their chronotype, and as the timing of sessions was not controlled for, we cannot examine the influence of the timing of each session. We intend to investigate these questions in our ongoing and future studies. We added this information to the Supplementary Methods section (SI, p. 2, l. 31-34):

"Participants were not instructed to follow a particular schedule before the experiment (in particular, our protocol did not involve a sleep restriction procedure). The timing of the experiment was determined by participants' availability and included both morning and afternoon sessions."

- Please specify each time you use 'slope' whether you mean downward or upward slope and if you report only one, why (no effect for the other)?

Response: Following the Reviewer's suggestion, we now always report both types of slope and clearly label each type (Upward Slope or U-Slope and Downward Slope or D-Slope).

- Were stimuli constantly presented or was there an interval between visual stimuli? If yes, how long was it?

Response: There was no interval between visual stimuli to accommodate the frequency tagging approach. We have added this information to the Online Methods (p. 28, l. 574-575):

"Face and digit stimuli were presented continuously, each stimulus appearing for a duration of 750 to 1250ms (random uniform jitter)."

- How many probes were collected per subject?

Response: A total of 60 probes (10 per experimental block, 30 per SART task) were collected. We made this clearer in the Online Methods (p. 29, l. 597-599):

"Each of the 12-15 min SART blocks included 10 interruptions (in total, 30 interruptions for each SART task and 60 interruptions per participant)."

- Was the slow wave detection performed after removing average voltage around entire window (64s), as specified in line 466?

Response: Yes. This step, coupled with the bandpass filtering, allows centring the EEG signal around 0. On a related note, we noticed and corrected an error in the setting of the bandpass filtering used for the slow wave detection (p. 31, l. 643).

6) Minor comments

- Line 456: Two electrodes were placed over the deltoid muscles to record electrocardiographic (ECG) activity. Do the authors mean electromyographic activity?

Response: No. This montage was indeed used as an alternative montage to record ECG activity. The classical ECG montage involves placing an electrode on the 5th intercostal space. This was impractical in some situations and, as ECG activity was not a main focus in this study, we opted for an alternative montage with two electrodes placed on the left and right deltoid muscles, which are easier to access.

- Supplementary table, line for Figure 4b: where does the mind state fit in? Instead of “Local sleep”, do the authors mean ‘mind state’ (as a predictor instead of local sleep)?

Response: We thank the Reviewer for spotting this error. For the new Figure 4, the predicted variables are the mental states and the predictors are slow-wave properties. As we are now reporting the results for all slow-wave properties (density, amplitude, downward and upward slopes), we updated the Table S1 accordingly.

- The asterisks in Figure 2 ^{***}, which refer to the likelihood ratio, are confusing, as one may erroneously interpret them as a difference between the ON and MB conditions

Response: We chose this display because this is typically used to show condition effects with ANOVAs. The legend indicates that: “In a-e, stars show the level of significance of the effect of mental states (^{***}: $p < 0.005$), which was obtained by the Likelihood Ratio Test (See Online Methods)”. We hope the legend makes the meaning of these stars unambiguous but we could remove the stars entirely if the Reviewer prefers.

- Figure 4: It is not immediately clear (even if there is a line separating the two rows) that the labels “MW vs ON” and “MB vs ON” refer only to the lower line. In addition the title ‘local sleep’ in Figure 4b is misleading, as this formulation is used to define the presence/absence of local sleep in Figure 4a, but in the lower line, slow waves parameters are shown instead. It would be better to more clearly separate the two panels, to indicate that only contrasts yielding significant differences are shown, and to replace ‘local sleep’ with ‘slow wave parameters’.

Response: We have now split Figure 4 in two new Figures. We also replaced ‘local sleep’ to ‘slow wave parameters’ to avoid equating the slow waves we detected in our data and the general concept of local sleep. These changes should address the Reviewer’s comment.

- The explanations on the Hierarchical Drift Diffusion Model are very difficult to understand, even after reading the online methods

Response: We revised the relevant part to improve the readability (Results, p. 16-17, l. 322-334):
“The DDM decomposes full reaction-time distributions and choice proportions into latent cognitive processes that are thought to underlie participants’ decisions in 2AFC tasks (see Online and Supplementary Methods and Figure S5). These include the time it takes for participants to start computing their response (Non-Decision Time or NDT [t]), the speed at which participants accumulate evidence for the two responses (drift rates for Go [v_{Go}] and NoGo [v_{NoGo}] responses), the amount of evidence needed to reach a decision (decision threshold [a]), the participants’ initial bias for one of the two responses (decision bias [z]) and the bias for the accumulation of evidence for one of the two responses (drift bias, [v_{Bias}]). A hierarchical Bayesian approach was used to fit the DDM to the reaction times obtained in the Go/NoGo tests⁴⁰ so that each parameter (v_{Go} , v_{NoGo} , a , z , t and v_{Bias}) was free to vary by participant, task and mental states (ON, MW and MB) or slow-wave occurrence (present vs. absent).”
We will also make all codes and data available upon publication to facilitate the replication of these results.
Raw data: https://osf.io/ey3ca/?view_only=680c39e7065649c3b783a4efec0a1a94
Code used for analyses: <https://github.com/andrillon/wanderIM>

- Figure 1, please explain how the slow waves were aligned (based on the first zero-crossing?)

Response: That is correct and we made this clearer in the manuscript (legend of Figure 3):
“Slow waves were aligned by their start, defined as the first zero-crossing before the negative peak (see Online Methods).”

REVIEWER COMMENTS

Reviewer #1 (Remarks to the Author):

I thank the authors for taking my comments seriously and in full, including both theoretical and analyses aspects. Looking forward to more work on the transitions of consciousness.

Reviewer #2 (Remarks to the Author):

The revisions to this paper have substantially improved the paper and its likely contribution to the question of the neural basis of different types of ongoing thought patterns. In particular, I thank the authors for shifting the emphasis to conclusions that are most closely aligned to the analyses they have performed as I believe this places their paper on a much more stable conceptual footing. Congratulations for such an excellent piece of work.

Signed

Jonathan Smallwood

Reviewer #3 (Remarks to the Author):

In response to my comments, the authors have clarified several methodological aspects.

The main issue, that the paper lacks a more direct proof for the fact that the slow waves they observe are sleep-related, remains. In the revised version the authors have replaced the term 'local sleep' with 'slow waves' in many places and now mention this limit in the discussion. Yet the study is still presented as a way to prove a hypothesis on local sleep (see for instance conclusions in the abstract) and given the methodological choices (targeted slow waves analysis), it makes no sense to reframe this study a different way. My concern is that this work will set a standard for future studies, making it legitimate to consider low-frequency oscillations in the wake EEG as sleep-related without more direct evidence.

My other concern is the methodological choice to present stimuli continuously, changing at very short intervals (approximately 1s), which makes it impossible to truly disentangle stimulus-related activity and spontaneously occurring slow waves (even if the average ERPs shown in the revised version are distinct from slow waves).

Dear Dr Andrillon,

Thank you again for submitting your manuscript "Wandering minds, sleepy brains: lapses of attention and slow waves in wakefulness" to Nature Communications. We have now received reports from three reviewers and, on the basis of their comments, we have decided to invite a revision of your work for further consideration in our journal. In particular, we require you to better address Reviewer 3's concerns, including clarifying throughout the text that the local sleep model is tentative and the circumstances under which this type of model would apply in a real-world setting.

When resubmitting, you must provide a point-by-point response to the reviewers' comments. Please show all changes in the manuscript text file with track changes or colour highlighting. If you are unable to address specific reviewer requests or find any points invalid, please explain why in the point-by-point response.

Important: In addition to the above, you must comply with the following editorial requests; we will not be able to proceed with your revised manuscript otherwise. Please also see the *Nature Communications* formatting instructions, which you may find useful while preparing your revised manuscript.

POLICIES AND FORMS REQUIRED FOR RESUBMISSION

* Please complete or update the following checklist(s) to verify compliance with our research ethics and data reporting standards. Address all points on the checklist, revising your manuscript in response to the points if needed.

The form(s) must be downloaded and completed in Adobe Reader rather than opened in a web browser. Each form must be uploaded as a Related Manuscript file at the time of resubmission.

Editorial policy checklist:

<https://www.nature.com/documents/nr-editorial-policy-checklist.pdf>

Reporting summary:

* Your paper uses custom code/software. Please complete the following code and software submission checklist and make your code available for reviewer assessment, if you have not already done so.

<https://www.nature.com/documents/nr-software-policy.pdf>

*We are currently changing our code sharing practices and now request authors to deposit code in a DOI-minted permanent repository (e.g. Zenodo through Github) and cite it in the main text. Please amend the text accordingly

<https://www.nature.com/nature-research/editorial-policies/reporting-standards#availability-of-computer-code>
<https://guides.github.com/activities/citable-code/>
<https://f1000research.com/articles/9-1257/v2>

DATA AND CODE AVAILABILITY

* All Nature Communications manuscripts must include a "Data Availability" section after the Methods section but before the References. If any of the data can only be shared on request or are subject to restrictions, please specify the reasons and explain how, when, and by whom the data can be accessed. For more information on this policy and a list of examples, see:

<https://www.nature.com/documents/nr-data-availability-statements-data-citations.pdf>

* Please also include a "Code Availability" section after the "Data Availability" section. If the code can only be shared on request, please specify the reasons. For more information on our code sharing policy and requirements, please see:

* We strongly encourage you to deposit all new data associated with the paper in a persistent repository where they can be freely and enduringly accessed. We recommend submitting the data to discipline-specific and community-recognized repositories; a list of repositories is provided

here: <http://www.nature.com/sdata/policies/repositories>

* To maximise the reproducibility of research data, we require you to provide a file containing the raw data underlying the following types of display items:

- Any reported means/averages in box plots, bar charts, and tables
- Dot plots/scatter plots, especially when there are overlapping points

- Line graphs

The data should be provided in a single Excel file with data for each figure/table in a separate sheet, or in multiple labelled files within a zipped folder. Name this file or folder 'Source Data', and include a brief description in your cover letter. The "Data Availability" section should also include the statement "Source data are provided with this paper."

To learn more about our motivation behind this policy, please see: <https://www.nature.com/articles/s41467-018-06012-8>

ORCID

* Nature Communications is committed to improving transparency in authorship. As part of our efforts in this direction, we are now requesting that all authors identified as 'corresponding author' create and link their Open Researcher and Contributor Identifier (ORCID) with their account on the Manuscript Tracking System prior to acceptance. ORCID helps the scientific community achieve unambiguous attribution of all scholarly contributions.

You can create and link your ORCID from the home page of the Manuscript Tracking System by clicking on 'Modify my Springer Nature account' and following these instructions. Please also inform all co-authors that they can add their ORCIDs to their accounts and that they must do so prior to acceptance.

If you experience problems in linking your ORCID, please contact the Platform Support Helpdesk.

HOW TO SUBMIT

Please use the link below to submit the following items as separate documents:

- Revised manuscript
- Any supplementary files
- Point-by-point response to the reviewers' comments, reproduced verbatim
- Cover letter to the editor
- Any completed checklist(s)

We would normally ask to see a revised version of this paper within three months but we appreciate revisions may take longer than usual and can extend this timeline if the Covid-19 pandemic prevents you from undertaking any further work for a longer period - please do get back to us on this nearer the time.

When evaluating your revised manuscript, we will not consider any similar papers published in the meantime to compromise the novelty of your study. See here for more information.

Best regards,

Henrietta

Henrietta Howells, PhD

Associate Editor
Nature Communications

REVIEWER COMMENTS

Note: We put excerpts of Reviewer 3's comments in bold fonts for emphasis.

Reviewer #1 (Remarks to the Author):

I thank the authors for taking my comments seriously and in full, including both theoretical and analyses aspects. Looking forward to more work on the transitions of consciousness.

Reviewer #2 (Remarks to the Author):

The revisions to this paper have substantially improved the paper and its likely contribution to the question of the neural basis of different types of ongoing thought patterns. In particular, I thank the authors for shifting the emphasis to conclusions that are most closely aligned to the analyses they have performed as I believe this places their paper on a much more stable conceptual footing. Congratulations for such an excellent piece of work.

Reviewer #3 (Remarks to the Author):

In response to my comments, the authors have clarified several methodological aspects.

The main issue, that the paper lacks a more direct proof for the fact that the slow waves they observe are sleep-related, remains. In the revised version the authors have replaced the term 'local sleep' with 'slow waves' in many places and now mention this limit in the discussion. Yet the study is still **presented as a way to prove a hypothesis on local sleep (see for instance conclusions in the abstract)** and given the methodological choices (targeted slow waves analysis), it makes no sense to reframe this study a different way. **My concern is that this work will set a standard for future studies, making it legitimate to consider low-frequency oscillations in the wake EEG as sleep-related without more direct evidence.**

In our previous rebuttal and in the revised version of the manuscript, we already acknowledged the reviewer's point that it would require intracranial recordings or a sleep deprivation approach to establish a direct proof of the link between wake and sleep slow waves. We admitted that this was not possible for this study. Thus, we instead relied on a converging array of evidence. We clearly outlined the limitations of this approach in the revised manuscript. To more directly reflect this, we modified the Discussion as follows (p. 26 | 535-546):

“Importantly, as our study builds on convergent lines of indirect evidence to argue that the slow waves we measured in waking participants are sleep-related, this interpretation is tentative and we do not wish to suggest that the presence of slow-frequency oscillations in wake EEG is by itself sufficient to establish a relation to sleep. Here, for example, we paired these observations with objective (pupil size) and subjective (vigilance ratings) markers of fatigue and checked that slow waves increased with the time spent on task. Future studies could use direct evidence from intracranial recordings or sleep deprivation to more solidly establish this interpretation. Intracranial recordings in particular can confirm whether wake slow waves are accompanied by episodes of neuronal silencing, as for sleep slow waves, and would help the identification of the exact neural mechanisms that generate the slow waves we report here.”

We also removed the mention of local sleep in the abstract and parts of the introduction to avoid the implication that “this study is presented as a way to prove a hypothesis on local sleep”. Finally, we linked slow waves to the broader literature on the neural correlates of the transition toward sleep, making our work less dependent on the local sleep hypothesis (p. 20-21 | 412-416). The manuscript should now reflect the novelty of our results above and beyond the local sleep interpretation.

My other concern is the methodological choice to present stimuli continuously, changing at very short intervals (approximately 1s), which makes it impossible to truly disentangle stimulus-related activity and spontaneously occurring slow waves (even if the average ERPs shown in the revised version are distinct from slow waves).

We opted for continuous stimulus presentation and did not include a resting-state session precisely because the aim of our current study was to investigate the relationship between slow waves, behavioural attentional lapses (e.g., slowing of RT, miss and false alarm) and phenomenological reports. We agree that including a resting state session could have allowed us to confirm that the observed slow waves also occur in the absence of stimuli or task. We now clearly outline this limitation in our manuscript (p. 26 l. 547-553):

“As our task involved the continuous presentation of visual stimuli, we could not fully disentangle the occurrence of slow waves and task-related events. Yet, we showed that slow waves differ from typical responses evoked by stimuli or participants’ responses (Figure S9). Future studies could include task-free resting state sessions to compare the occurrence and properties of slow waves occurring during a task and without a task. Relatedly, further investigation could help determine to which extent our findings generalize to everyday situations (e.g., driving, attending a lecture, reading, etc).”

REVIEWER COMMENTS

Reviewer #3 (Remarks to the Author):

Thank you for considering my comments.

REVIEWER COMMENTS

Reviewer #3 (Remarks to the Author):

Thank you for considering my comments.

We hope the latest version of the manuscript is improved.